



# The role of North Atlantic-European weather regimes in the surface impact of sudden stratospheric warming events

Daniela I.V. Domeisen[1], Christian M. Grams[2], and Lukas Papritz[1]

[1]Institute for Atmospheric and Climate Science, ETH Zürich, Switzerland
[2]Institute of Meteorology and Climate Research (IMK-TRO), Department Troposphere Research, Karlsruhe Institute of Technology (KIT), Karlsruhe, Germany

**Correspondence:** Daniela I.V. Domeisen (daniela.domeisen@env.ethz.ch)

**Abstract.** Sudden stratospheric warming (SSW) events can significantly impact tropospheric weather for a period of several weeks, in particular over the North Atlantic and Europe. However, not all SSW events exhibit the same tropospheric response, if any, and it remains an open question what determines the existence, location, timing, and strength of the downward impact. We here explore the role of the state of the tropospheric flow in the North Atlantic region at the onset of SSW events for

determining the subsequent surface impact. A refined definition of seven North Atlantic tropospheric weather regimes indicates the Greenland blocking (GL) and Atlantic Trough (AT) regimes as the most frequent large-scale patterns following the weeks after an SSW. While GL is dominated by high pressure over Greenland, AT is dominated by a southeastward shifted storm track in the North Atlantic. We find that a blocking situation over western Europe and the North Sea (European Blocking) at the time of the SSW onset favours the GL response and the associated cold conditions over Europe. In contrast, an AT response

and mild conditions are more likely if GL occurs already at SSW onset. For the remaining tropospheric flow regimes during SSW onset, we find no clear response. The results indicate that the tropospheric impact of SSW events critically depends on the tropospheric state during the onset of the SSW, which could provide crucial guidance for subseasonal prediction.

## 1  Introduction

Sudden stratospheric warming events can have a significant impact on the tropospheric large-scale circulation and hence on surface weather (Baldwin and Dunkerton, 2001). However, a robust detection and quantification of the downward impact of SSWs remains challenging. First of all, the number of SSWs in the record of satellite-era reanalysis is small (26 events from 1979 - 2019), while the case-to-case variability is large. Second, the internal variability of the troposphere itself is high, such that it can mask a stratospheric influence. Predicting if, when, and where a downward impact from SSW events will occur is

therefore not straightforward, yet a better prediction of the type and timing of a downward impact would significantly benefit a wide range of users.





The downward impact is communicated by a range of mechanisms including synoptic- and planetary-scale waves (e.g. Song and Robinson, 2004; Domeisen et al., 2013; Hitchcock and Simpson, 2014; Smith and Scott, 2016). In particular, after SSW events the North Atlantic-European region (NAE) tends to exhibit more persistent states of the negative phase of the North

Atlantic Oscillation (NAO-, Domeisen, 2019), as well as more frequent transitions towards NAO- and fewer away from NAO- (Charlton-Perez et al., 2018). NAO- is associated with enhanced meridional air mass exchanges, in particular, more cold air outbreaks in Northern Europe but fewer over the Nordic Seas (Kolstad et al., 2010; Kretschmer et al., 2018b; Papritz and Grams, 2018; Huang and Tian, 2019), as well as anomalous precipitation in Southern Europe (Butler et al., 2017; Ayarzagüena et al., 2018). The Pacific sector tends to be less strongly affected in the aftermath of SSW events (Greatbatch et al., 2012; Butler

et al., 2017), though the occurrence of wave reflection in the stratosphere can be associated with Pacific blocking (Kodera et al., 2016) and cold spells over North America (Kretschmer et al., 2018a). Given the preferred occurrence and the increased persistence of certain surface signatures after SSW events as compared to climatology, medium- to long-range predictability over Europe has been suggested to increase after SSW events (Sigmond et al., 2013; Domeisen et al., 2015; Karpechko, 2015; Butler et al., 2016; Scaife et al., 2016; Jia et al., 2017; Beerli et al., 2017; Butler et al., 2019).

Despite the preferred occurrence of the negative phase of the NAO, the downward influence of an SSW event on the evolution of the tropospheric flow can be highly variable between events. This issue is further complicated by the fact that there exists a range of different metrics for characterizing the downward impact, with each definition yielding a different set and number of SSW events with apparent surface impacts. Most definitions of a downward impact are based on large-scale circulation indices such as the NAO, the Northern Annular Mode (NAM), or the Arctic Oscillation (AO) (Karpechko et al., 2017; Charlton-Perez

et al., 2018; Domeisen, 2019). Depending on the definition, between one to two thirds of SSW events exhibit a downward impact in the NAE region (Karpechko et al., 2017; Charlton-Perez et al., 2018; Domeisen, 2019).

While a causal downward link from the stratosphere after SSW events has been confirmed (e.g. Gerber et al., 2009), remote forcings can affect both the stratosphere and the troposphere, and thereby either mask or strengthen the downward response from the stratosphere. Indeed, a range of tropical remote connections can impact the NAE region through both a tropospheric

and a stratospheric pathway (Attard et al., 2019), such as the Quasi-Biennial Oscillation (QBO) (Gray et al., 2018; Andrews et al., 2019), the MJO (Garfinkel et al., 2014; Barnes et al., 2019), and the El Niño Southern Oscillation (ENSO) (Jiménez-Esteve and Domeisen, 2018; Domeisen et al., 2019b), in addition to extratropical tropospheric forcing in the North Pacific (Honda and Nakamura, 2001; Sun and Tan, 2013; Drouard et al., 2013), Arctic sea ice (Sun et al., 2015), and snow cover in Eurasia (Cohen et al., 2014). It therefore has to be kept in mind that the stratosphere is often only one possible forcing of the

troposphere.

In addition, it has recently been suggested that the precursors to SSW events with a downward influence differ from those without such a tropospheric impact in terms of strength and location (Domeisen, 2019; Zhang et al., 2019), in particular with respect to forcing over Eurasia (White et al., 2019; Tyrrell et al., 2019; Peings, 2019). Given the large tropospheric internal variability and the influence of other remote effects mentioned above, it appears plausible that also the tropospheric state at

the time of occurrence of an SSW plays an important role in shaping the characteristics of its downward impact. For example, tropospheric jet characteristics have been suggested to affect the downward impact of SSW events (Chan and Plumb, 2009;



Garfinkel et al., 2013). The goal of this study is to further elucidate the role of the troposphere in the downward impact of SSW events based on the observational record. More specifically, we here address the following questions:

1. Is the tropospheric flow evolution in the NAE region after an SSW statistically different from that without an SSW?

2. Are the characteristics of the tropospheric flow response to an SSW dependent on the weather regime in the NAE region at the onset of the SSW?

To address these questions, we focus on the dynamical variability during SSW events using a classification of the large-scale flow in the Euro-Atlantic sector into seven weather regimes. This expands on Charlton-Perez et al. (2018), who have used four weather regimes (NAO+, NAO-, Scandinavian blocking and Atlantic ridge) and characterized changes in the transition probability between regimes in the aftermath of SSW events. Prior work based on this extended regime definition revealed important differences in the surface weather response to the state of the stratosphere, which remain hidden using the canonical four NAE regimes (Papritz and Grams, 2018; Beerli and Grams, 2019). Here, we focus on the evolution and surface impact of tropospheric anomalies as a function of the weather regime present at the onset of the SSW event. Furthermore, we discuss several case studies, including the different surface impacts of the 2018 and 2019 SSW events.

## 2 Data and Methods

### 2.1 Data and Classifications

ERA-interim reanalysis (Dee et al., 2011) from 1979 to present is used for all figures. The SSW central dates are identified as the day of reversal of the zonal mean zonal winds to easterly at 10 hPa and 60°N in midwinter (Nov - Mar) for ERA-Interim (1979 - 2019), yielding 26 SSW events for the period 1979-2019. For the period 1979 to 2013, these reversal dates are given in Table 2 in Butler et al. (2017). The SSW central dates for the remaining years are 12-Feb-2018 and 02-Jan-2019.

The tropospheric flow over the NAE region is described in terms of quasi-stationary large-scale flow patterns, given by seven year-round weather regimes defined in Grams et al. (2017) based on six-hourly data for the period 1979-2015. We use this weather regime classification to stratify SSWs according to the large-scale tropospheric flow conditions at their onset. To do so, we select for each SSW the dominant weather regime active during at least one 6-hourly timestep throughout the day of an SSW. As for the canonical seasonal definition using four regimes (e.g. Michelangeli et al., 1995; Michel and Rivière, 2011; Ferranti et al., 2015; Charlton-Perez et al., 2018), the mean patterns of the seven regimes are based on a k-means clustering in the phase space spanned by the leading seven empirical orthogonal functions (EOFs; explaining 76% of the variance) of 10-day low-pass filtered 500hPa geopotential height anomalies. In addition, a weather regime index following Michel and Rivière (2011) is employed for defining objective weather regime life cycles and for a filtering of timesteps without a clear regime structure ("no regime" category). This life cycle definition allows for a continuous extension of the weather regime attribution to more recent data without repeating the EOF analysis and clustering (here done for the years 2016-2019).

Three of the seven regimes are dominated by a cyclonic 500 hPa geopotential height anomaly ("cyclonic regimes"; cf. Figs. A1a-c): the Atlantic Trough (AT) regime with cyclonic activity shifted towards western Europe, the Zonal regime (ZO), and the





Scandinavian Trough (ScTr) regime. The remaining four regimes are dominated by a positive geopotential height anomaly and

are referred to as "blocked regimes" (Figs. A1d-g): Atlantic Ridge (AR), European Blocking (EuBL), Scandinavian Blocking (ScBL), and Greenland Blocking (GL).

The frequency of occurrence of the seven regimes is modulated by the stratosphere, which can be understood in terms of the link between the respective regimes and the NAO (Beerli and Grams, 2019, their Figs. 2,6) and the link between the stratospheric polar vortex and the NAO (Charlton-Perez et al., 2018). Since ZO and ScTr project onto NAO+, they are suppressed

after a weakening of the stratospheric polar vortex, and vice versa after a strengthening. In contrast, GL strongly projects onto NAO- and is enhanced following a weak stratospheric polar vortex, while it is suppressed in the aftermath of a strong vortex. EuBL and AT do not project strongly onto either NAO phase and are, thus, only weakly modulated by the strength of the stratospheric polar vortex.

## 2.2 Statistical testing

The rare occurrence of SSWs (26 events during 1979-2019) and the subsequent stratification according to tropospheric flow conditions requires careful statistical testing to extract significant and robust results that are distinct from sampling uncertainty. In the following, we outline our general approach for statistical testing consisting of two steps. The specific tests are then further detailed when describing the respective analyses. First, we estimate confidence intervals by randomly resampling 100 times the original sample with repetition. The number of random samples is chosen according to the maximum number of

possible combinations with repetitions of the smallest subset of SSW events that will be considered in this study ($N = 5$ events corresponding to 126 independent combinations). Anomalies are considered robust if the width of the confidence interval is smaller than the amplitude of the anomaly. Second, we test for significance by computing 1000 Monte-Carlo samples by drawing random timesteps from the entire period. We then compare the Monte-Carlo distribution of sample means with the confidence intervals of the original sample. The sample mean is significant at, e.g., the 10% level, if the confidence intervals

overlap by less than 10% with the Monte Carlo distribution.

## 3 Weather regimes during SSW events

As a first step, we evaluate the sequence of weather regimes from 60 days before to 60 days after an SSW for all 26 SSW cases during 1979-2019 (Fig. 1). This suggests a preferred occurrence of AT (purple) and GL (blue) during the weeks after an SSW compared to the weeks before. Likewise, AR (yellow) and the related ScTr (orange) regimes are more frequent in the period

1-3 weeks before the SSW onset.

The 5-day running mean frequency of weather regimes around SSW events provides a more complete overview over the modulation of regime frequencies after SSWs (Figure 2). In addition, we show deviations of weather regime frequencies from climatology in Fig. A2. For testing significance, we compute the distribution of lagged 5-day mean frequencies in 1000 random samples obtained by selecting for each day in the original sample a random day ±15 days around the original day of year but

from a different winter. In addition, the random day must exhibit the same weather regime as the original day to replicate





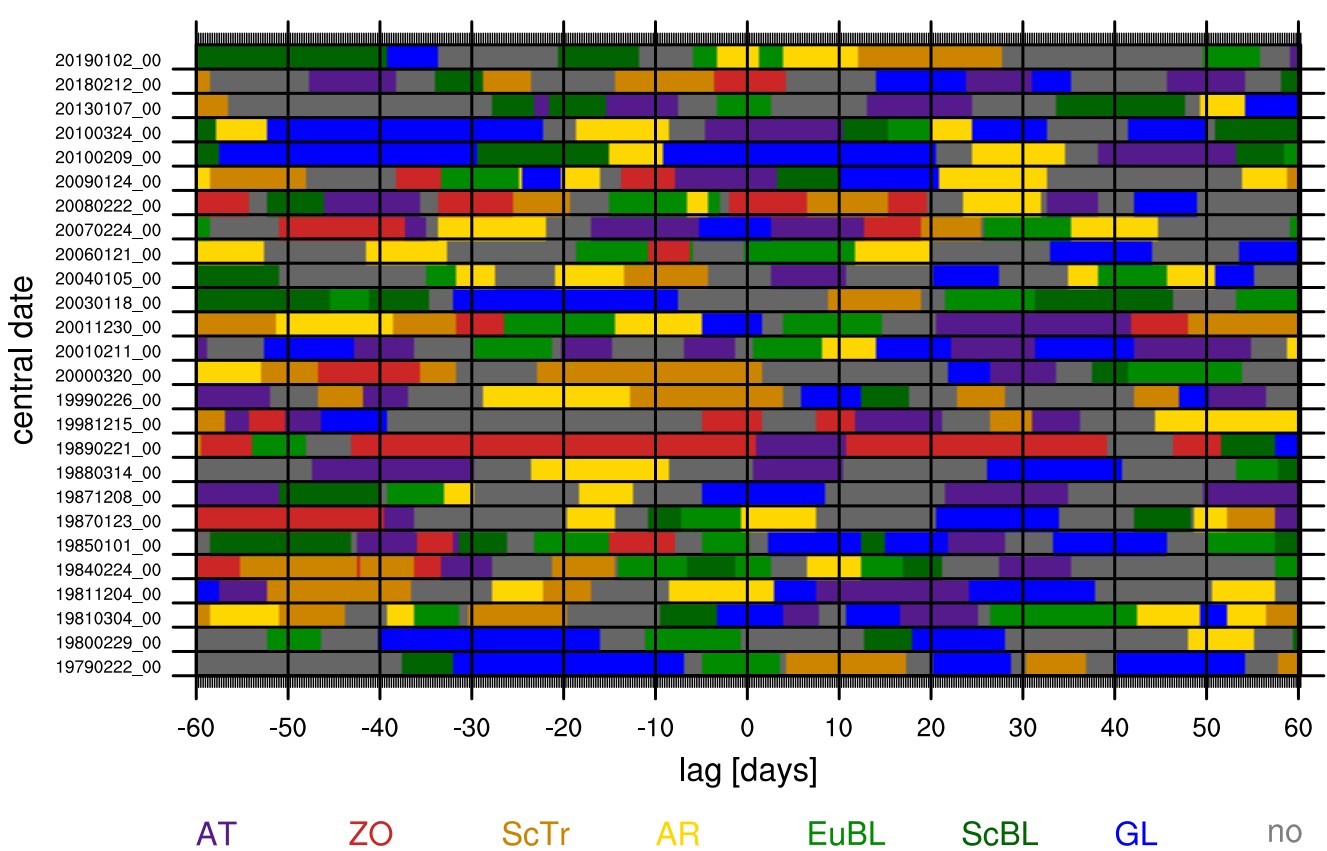

**Figure 1.** The sequence of the dominant weather regimes (colors indicated in legend) for -60 to +60 days with respect to the onset of each of the 26 SSW events (lag 0) between 1979 and 2019. The central dates of the SSW events are indicated on the left.

potential regime-dependence. We then compute the mean lagged weather regime frequency for each random sample as for the original sample and test for significance at the 10% level (bold).

Figure 2a indicates the absolute 5-day mean frequency of weather regimes in the NAE sector for a composite of all SSW events. GL and EuBL are the most prominent regimes at the onset of SSW events with a relative 5-day mean frequency of

more than 18% and 20%, respectively. Thereby the frequency of EuBL is significantly enhanced from 5 days prior until the onset of the SSW, in agreement with Woollings et al. (2010) and Nishii et al. (2011). The cyclonic regimes ZO, ScTr, as well as the blocked regimes AR and ScBL tend to be suppressed at the time of SSW events, consistent with the latter regimes' strong projection onto NAO+, which also tends to be suppressed after SSW events (Charlton-Perez et al., 2018). However, AR is significantly more prominent around 15 days before the onset of an SSW event, which agrees with the suggested precursor role

of blocking over the Atlantic before SSW events (Martius et al., 2009). After the SSW onset, AT frequencies are significantly enhanced, peaking at more than 23% after 7 days. Thereafter, GL (lag 12 to 42 days) and AT (lag 17-35 days) are the most likely weather regimes (see also Fig. A2a), but frequencies for both are only around 20-25% and none of the two clearly dominates.



This behaviour obscures the potential tropospheric impact of an SSW in a composite as AT and GL trigger contrasting large-scale weather conditions (rather mild and windy for AT, cold and calm for GL) for large parts of Europe (Beerli and Grams,
135  2019).

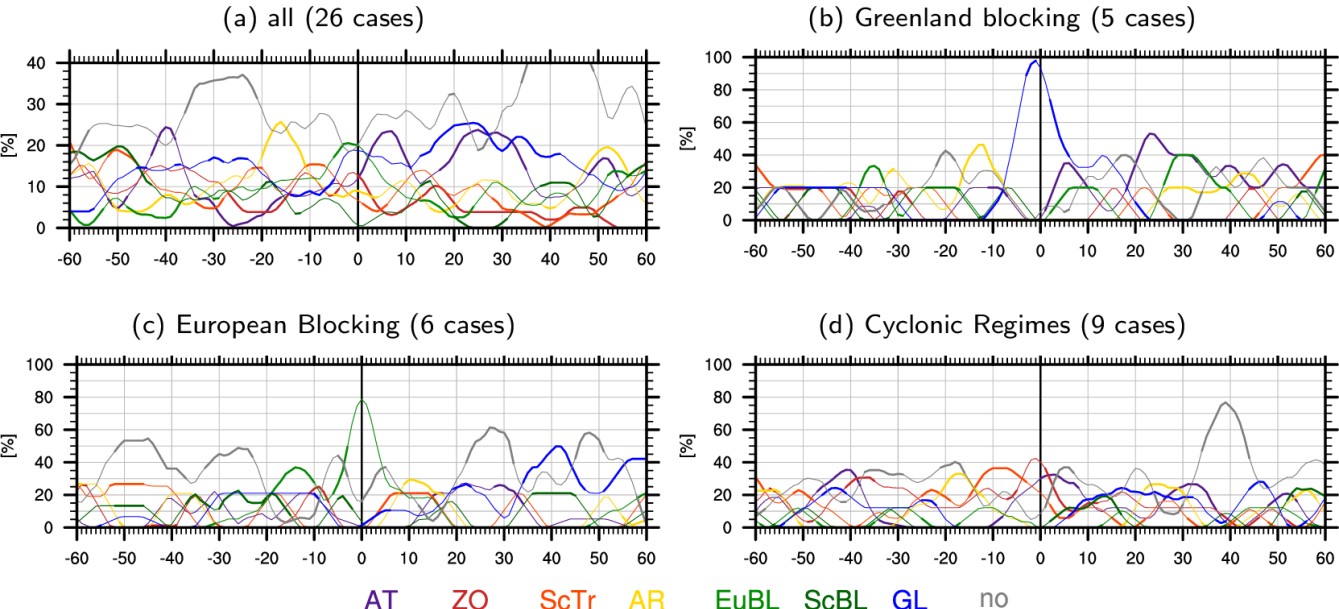

**Figure 2.** 5-day running mean frequency of weather regimes centred on the onset of the SSW event (lag 0) for (a) all SSW events and (b-d) conditional on the dominant weather regime at lag 0: (b) Greenland blocking, (c) European blocking, and (d) cyclonic regimes (ZO, AT, and ScTr). The 5-day mean frequencies are computed from 6-hourly weather regime data from lag -60 days to lag 60 days. Note the different y-axis in (a). The bold parts of the lines indicate significant deviations from climatology (see text for details).

We now sub-divide the 26 SSW events with respect to the weather regime that dominates during the SSW onset: GL (5 cases), EuBL (6 cases), and the cyclonic regimes (ZO, ScTr, AT; 9 cases). The remaining 6 cases either have no clear regime signature (no-regime, 3 events) or are associated with AR (3 events) at their onset. Because of the small sample size, we do not consider these cases here. For the GL subset (Fig. 2b / A2b), all other regimes are subsequently suppressed except for AT
and EuBL. The frequency of GL itself drops immediately after the SSW to below 10% around a lag of 20 days – far below its climatological mean frequency. AT, and to a lesser degree also EuBL, become significantly more frequent immediately after the SSW until about a lag of 10 days, reaching absolute frequencies of 35% and 20%, respectively. After a period with no clear regime assignment, AT becomes the dominant regime starting at lag 18 days and peaking above 50% absolute frequency about 23 days after the SSW, thus remaining significantly enhanced until a lag of 33 days. From lag 25 days until lag 40 days, EuBL
becomes significantly enhanced peaking at 40% absolute frequency around lag 30 days.

For the EuBL subset (Fig. 2c / A2c), the subsequent regime frequencies are quite different to GL at the onset of an SSW. First, the frequencies of ScTr and AR are significantly enhanced directly after the SSW, with peaks at 20% and 30% absolute frequency at lag 10 days. This is then followed by a period of preferred occurrence of GL (around lag 20 days) and AT (around





lag 30 days) with a frequency of about 25% each. The dominance of GL from lag 35 to 45 days (50% peak frequency) is
particularly striking. At the same time, also ScBL is enhanced with a frequency of 20%, while all other regimes are suppressed.

Cyclonic regimes at the time of the SSW (Fig. 2d / A2d) exhibit a less prominent regime frequency modulation after an SSW
compared to the EuBL and GL subsets. Still, GL (lag 10-35 days, lag 45 days), AR (lag 20-30 days), and AT (lag 25-35 days)
are significantly enhanced, but absolute frequencies barely exceed 20%. Most often no single regime dominates after an SSW
event with a cyclonic regime at lag 0, hinting at cases with a "missing" response after the SSW event.

Despite the large tropospheric variability in the aftermath of SSW events, the investigation of lagged regime frequencies
revealed that (1) the AT and GL regimes are more likely to follow an SSW (as compared to other weather regimes) and (2)
that this subsequent modulation is sensitive to the tropospheric flow regime at the onset of the SSW. The dominance of either
EuBL or GL at the time of the SSW onset strongly influences the further development of the tropospheric regimes, i.e. towards
a significantly more likely GL response (after EuBL at lag 0) vs. AT (after GL at lag 0) 3-4 weeks after an SSW. This indicates
that the presence of a particular tropospheric regime at the onset of the SSW can be responsible for the subsequent evolution
of the tropospheric flow, and hence, for the surface impact of the SSW and the associated surface weather.

## 4   Temporal Evolution of the Downward Impact

Given the strong influence of the tropospheric state at the time of the onset of an SSW on the weather regime frequencies in
the subsequent days, we focus in the following on the modulation of the stratosphere-troposphere coupling for the previously
discussed sets of SSWs. For that purpose, we evaluate the temporal evolution of standardized geopotential height anomalies
averaged over the NAE sector (-80°E to 40°E / 60°N to 90°N) by compositing a given set of SSW events. Using the full
hemisphere, that is, the full longitude range instead of the here used sectorial view over the North Atlantic, yields the same
qualitative results due to the strong imprint of the anomalies induced by the SSW in the NAE sector (Fig. A3).

In order to evaluate the robustness of the composites, we compute confidence intervals by resampling the SSW events 100
times with repetition. If the magnitude of the anomaly in the actual composite exceeds the inter-quartile or the $10^{th}$-$90^{th}$
percentile range of the confidence distributions, the anomaly is considered robust or highly robust, respectively (see Fig. A4).
To test for statistical significance, we further pick 1000 samples of N randomly chosen days among all winter days (December
to March 1979 to 2019) within a given set of tropospheric weather regimes. Thereby, we exclude days that are within ±60 days
of the onset of an SSW event to ensure that the reference samples are independent of the SSWs. For each of these samples we
then compute lag composites, yielding a distribution of the anomalies obtained from a random sampling. If these distributions
overlap by less than 25 % or 10 % with the confidence distributions, the composite anomalies are unlikely or highly unlikely,
respectively, obtained from random samples.

Compositing all SSW events (Fig. 3a) yields the classical dripping paint plot of Baldwin and Dunkerton (2001, their Fig. 2).
The qualitative differences to the figure from Baldwin and Dunkerton (2001) are likely due to the different variable (geopo-
tential height in our study vs NAM), the number of events (26 in our study vs 18) in a different time period (1979-2019 in our
study vs 1958-1999). When compositing all SSW events, the downward impact between 10 to 60 days after the SSW onset



is robust (but not highly robust according to our definition, see Fig. A4a). Together with the relatively weak amplitude of the anomalies, this reflects the rather large case-to-case variability in the tropospheric impact of SSWs. A highly robust anomaly can only be observed at around 15 days after an SSW. Yet, this anomaly is rather unlikely obtained from a random sampling

as evident from the less than 25 % overlap between the confidence and random distributions (Fig. 3a). This suggests that in the aftermath of an SSW, indeed positive geopotential height anomalies over the NAE sector are more likely than in the absence of an SSW.

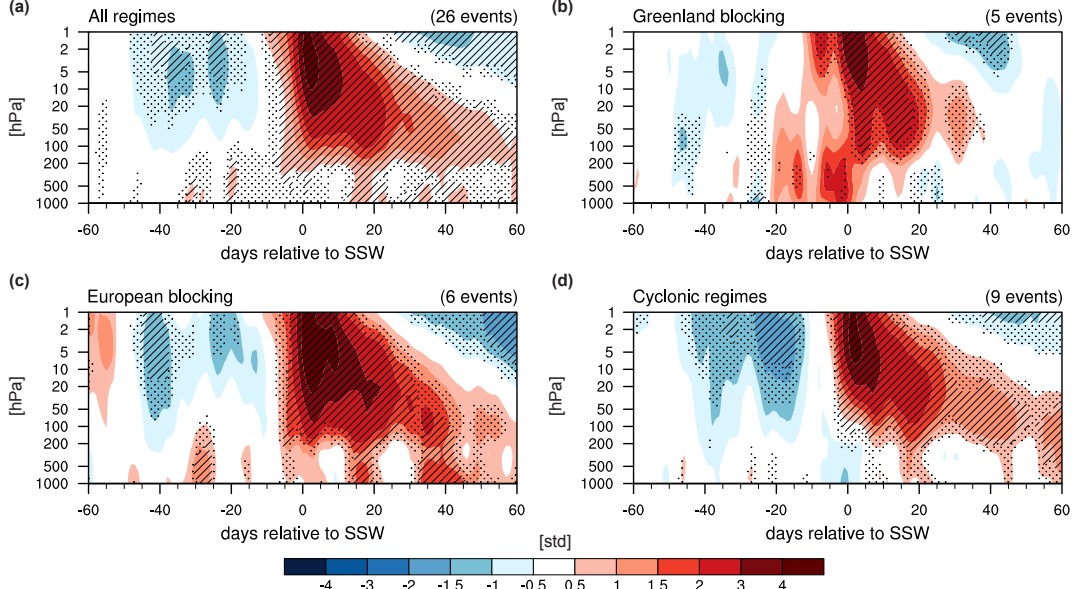

**Figure 3.** Standardized geopotential height anomalies for the sector -80°E to 40°E / 60°N to 90°N for (a) all SSW events, and (b - d) sub-divided by the weather regime that is dominant at the onset of the SSW event as indicated in the panel titles. Hatching (stippling) indicates that the confidence intervals and the random distributions overlap by less than 25% (10%).

SSW events that occur during GL (Fig. 3b) are associated with an immediate, strongly positive anomaly in the troposphere. Consistent with Fig. 2b, when GL is present at the onset of the SSW, GL or AR are often already present before the SSW event,

which is likely the cause of the positive tropospheric geopotential height anomalies several days prior to the event. Notably, there are no significant and robust (cf. Figs. 3b and A4b) anomalies after 10 days of the onset of the SSW except for a weak negative geopotential height anomaly after 20 days, indicating a cyclonic flow regime in the NAE region. This is consistent with the significantly enhanced likelihood for the occurrence of the AT regime at this lag (Fig. 2b). The immediate positive geopotential height anomalies and the weak tropospheric response in the aftermath of the event are archetypal for SSWs with

GL at the onset and not the result of cancellations in the composites. Indeed, they are also evident for individual SSWs, such as the SSW on 8-Dec-1987 that exhibited a dominant GL regime for an extended period around the SSW onset (Figs. 4a and 6a).

For EuBL at the onset of the SSW event, only a weakly significant positive anomaly can be observed at the time of the SSW, but highly robust, significant, and strongly positive geopotential height anomalies are present in the troposphere at lags of 15



- 20 and 30 - 55 days after the SSW event (Fig. 3c), as exemplified by the presence of GL in the aftermath of the SSW on

11-Feb-2001 (Figs. 4b and 6b). These positive anomalies are consistent with the finding that first AR and then GL are much more likely in the aftermath of an SSW with EuBL at lag 0 (compare to Fig. 2c). Furthermore, comparing to the panel for all SSW events (Fig. 3a) indicates that the EuBL cases dominate the perceived downward response in the canonical response for SSW events.

During cyclonic regimes at the onset of the SSW, there is no substantial tropospheric anomaly in the NAE region at the time

of the SSW, but a positive albeit weak anomaly can be observed around days 15 - 20 after the SSW event (Fig. 3d), which is again illustrated by a case study of the SSW events on Feb 12, 2018 (Fig. 4c). This anomaly is not highly robust, but it is nevertheless significantly different from a random sample at the 25 % level. Several SSWs with a cyclonic regime at the onset are followed by GL at a longer lag (Fig. 2d), thus, likely causing these anomalies. Still the GL frequencies only reach 25% at most and also other regimes occur more often albeit with low frequencies around 25%. These findings and the small amplitude

of the anomalies suggest that the variability in the tropospheric response to SSWs is large after a cyclonic regime at lag 0, which is also confirmed by the inspection of individual cases (not shown).

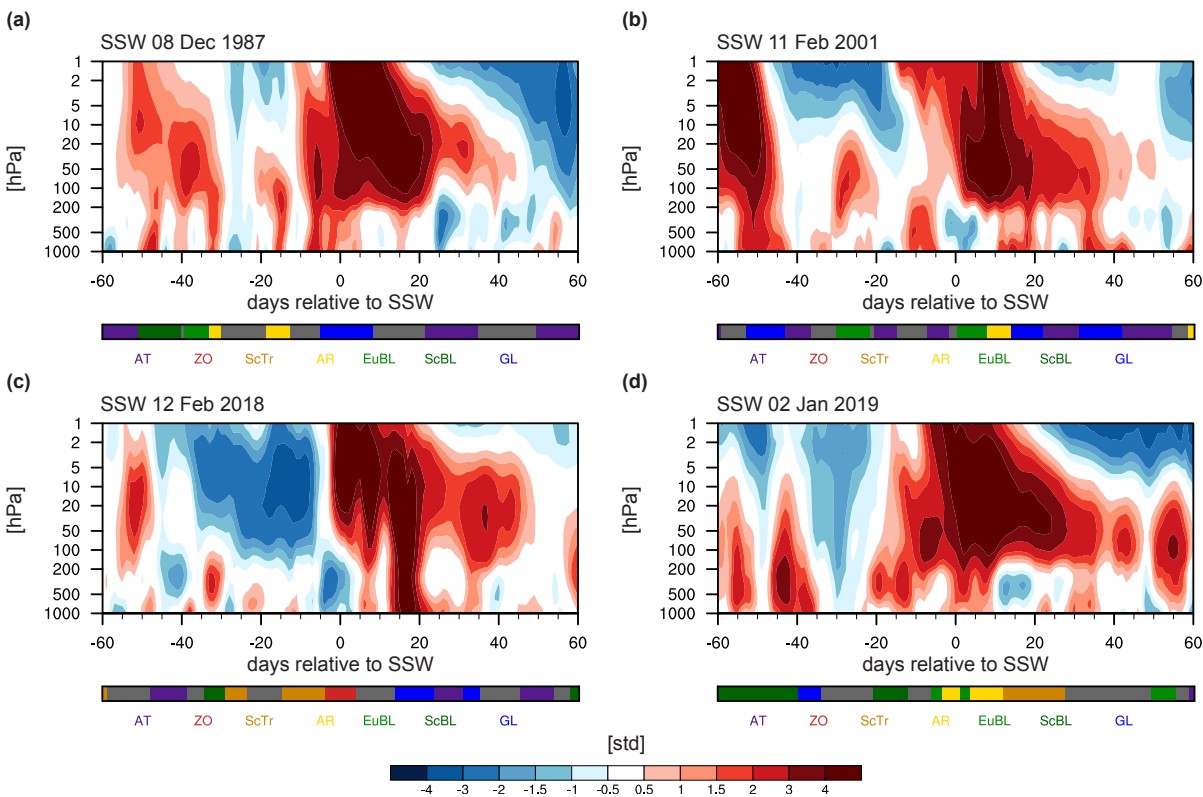

**Figure 4.** As Figure 3 but for individual SSW events on (a) 8-Dec-1987, (b) 11-Feb-2001, (c) 12-Feb-2018, (d) 2-Jan-2019. Dominant weather regimes are indicated by the colors at the bottom of each panel, and the color code is indicated in the legend.



## 5   Impact on Surface Weather

Since each weather regime is associated with characteristic surface weather, the modulation of regime successions in the aftermath of an SSW by the tropospheric state at the time of an SSW is likely a key contributor to the marked variability in the surface impact. Hence, we here consider spatial composites of 2m temperature anomalies and anomalies of 500 hPa geopotential height (Z500') for the three groups of SSW events discussed in the previous sections (Fig. 5, Fig. A5). The response in the composite of all SSW events is shown for comparison in Fig. A5.

During SSWs dominated by GL at the onset, initially strong warm anomalies prevail over Greenland and the Canadian Archipelago, whereas western Russia and Scandinavia are anomalously cold, consistent with the anomalous ridge over Greenland and the low geopotential height anomalies over Scandinavia (Fig. 5a). Consistent with the subsequent progression of weather regimes - typically towards the cyclonic AT regime or EuBL - mild conditions are established throughout central Europe (from a lag of 20 days onwards) and later shifting towards Scandinavia. This is in stark contrast to the negative NAO and the associated cold conditions that are commonly expected as the canonical response to SSWs over Europe (Butler et al., 2017; Kolstad et al., 2010; Domeisen et al., 2019a).

In the case of SSWs with EuBL at the onset, cold anomalies prevail over Northern Europe, albeit also extending over large parts of central Europe (Fig. 5b). They peak at -4 K to -6 K around lags of 20 and 40 days, which corresponds well with the occurrences of the GL regime. Note that cold anomalies for the canonical response of all SSWs are much weaker (cf. Fig. A5) reaching -1 K to -3 K in Central Europe. The regression of initial positive Z500' over the eastern North Atlantic to Greenland along with a strengthening of negative Z500' over the southeastern North Atlantic around lag 15-30 days is striking. Furthermore, the two occurrences of GL are associated with warm anomalies over Greenland and North America. Finally, as expected by the varied regime succession for the SSWs with cyclonic regimes at their onset, composite temperature and Z500' are weaker (Fig. 5c). As a result of these contrasting surface impacts for the different categories, the temperature anomalies when all SSWs are taken together remain relatively weak in comparison (Figure A5).

## 6   SSW Case Studies for 2018 and 2019

Due to the small number of SSW events in the observational record, we here in addition present selected case studies of SSW events. Given the recent discussion concerning the 2018 and 2019 SSW events and their respective downward impacts (Ayarzagüena et al., 2018; Karpechko et al., 2018; Rao et al., 2019; Kautz et al., 2019), we will in the following focus on these. The SSWs in 2018 and 2019 were both classified as split events, i.e. the stratospheric polar vortex split into two smaller daughter vortices. 2018 was a "classic" split, where two anticyclones from the Aleutian islands and from Northern Europe moved poleward and split the vortex. In 2019, a single anticyclone moved poleward in the stratosphere from the Aleutian islands and split the vortex. For both events, the cyclonic anomalies associated with the daughter vortices at 10hPa then moved over Northern Europe and North America in the days following the onset of the SSW. For the 2018 event, the cyclonic anomaly over Northern Europe then weakened, yielding a pattern of a strong cyclone over Canada and a strong anticyclone connecting Siberia and Greenland around 6 days after the SSW. For the 2019 event, the cyclonic anomalies remained very persistent for

**Figure 5.** Surface impact for SSWs with (a) Greenland blocking, (b) European blocking, and (c) cyclonic regimes at the onset. Shading indicates composite 2m temperature anomalies, black contours correspond to geopotential height anomalies at 500 hPa at intervals of 50 hPa with negative values dashed. Fields are averaged over 5 days between lags 0 to 50 days.





about a month over Canada and Siberia, with an extension towards Europe. Despite the similar classification in the stratosphere there were strong differences in the surface impact of the 2018 and 2019 SSW events.

At the time of the 2018 SSW event, the troposphere over the North Atlantic was in a cyclonic regime (zonal, i.e., related to NAO+, Fig. 4c). The troposphere subsequently transitioned into a weak European and then Scandinavian blocking a few days after the onset of the event, though none of these regimes was persistent enough to be picked up by the 5-day persistence

criterion applied to the weather regimes in this analysis (cf. Fig. 6c). The block then retrograded towards Greenland, indicated by the weather regime analysis as a significant anomaly around a lag of 14 days after the SSW onset. The troposphere subsequently remained in the Greenland blocking (NAO-) regime for around 20 days, with a significant projection onto an Atlantic trough, though the projection onto GL remains high throughout this period (cf. Fig. 6c and A6a). This transition is also documented in Ayarzagüena et al. (2018) and can clearly be seen in Figures 4c and A6a, which shows a weak downward impact

during the EuBL / ScBL regime, and a strong downward impact during GL / NAO-. Consistent with Fig. 3d, the strongest downward impact after the cyclonic regime at SSW onset occurred around lags of 15 to 25 days after the SSW onset.

For the SSW event on January 2, 2019, the tropospheric situation was significantly different from the 2018 SSW event and highly anomalous in comparison to other SSW events, as can be seen from comparing 2019 to the other SSWs in Figure 1. The regime around the onset of the SSW was dominated by an Atlantic ridge and European blocking (cf. Fig. 6d and A6b). The

subsequent response is atypical for the period following SSW events (i.e., only very few SSWs are followed by an Atlantic ridge, see Figure 1). The response then transitioned to a persistent Scandinavian trough for an extended period from lags 10 to 25 days after the SSW event (Fig. 4d and A6b). The ScTr regime tends to be associated with anomalously northwesterly flow into Central Europe (Grams et al., 2017) and was responsible for high snowfall in the European Alps.

## 7  Summary and Discussion

In this study we have investigated the tropospheric response in the NAE region in the aftermath of sudden stratospheric warming events and its dependence on the tropospheric weather regime at the onset of the SSW. To that end, we have exploited in a statistical framework the observational record of the satellite era (1979 - 2019) as represented in the ERA-Interim reanalysis. Our conclusions are as follows:

1. In the aftermath of an SSW event, the tropospheric flow in the NAE region exhibits an evolution that is unlikely to
occur in the absence of an SSW. Specifically, positive geopotential height anomalies related to Greenland blocking are statistically more likely to occur after the onset of the SSW than in the absence of an SSW. This is consistent with the expected (canonical) response of the troposphere to SSWs (e.g. Charlton-Perez et al., 2018).

2. The highly significant and highly robust positive geopotential height anomalies found in the period 10-60 days after SSWs are predominantly the result of SSWs with European blocking at their onset. This is manifest in a transition
from EuBL to GL that then dominates at lags of 15-20 and 30-55 days after the SSW onset, which is statistically



significantly different from the natural progression from EuBL to GL. For other tropospheric regimes at the SSW onset the tropospheric response is weaker and neither highly significant nor highly robust.

3. SSWs that occurs during cyclonic weather regimes exhibit a considerably weaker and less significant response with a modestly enhanced likelihood for GL. For GL at the onset, a weak preference for cyclonic flow regimes around 20-30 days after the SSW is apparent, with a much weaker and less persistent surface response in the aftermath of the SSW as compared to SSW onsets dominated by EuBL.

4. Given the clear influence of the regime present at the onset of an SSW event on the subsequent evolution of the tropospheric flow, different surface signatures result. Specifically, the canonical signature in 2m temperature, e.g., cold conditions prevailing over much of northern Europe, occurs for the EuBL cases. In contrast, mild temperatures in large parts of Europe are found for SSWs with GL at their onset. It is important to distinguish these cases, since although EuBL and GL frequently occur at the onset of SSW events, they lead to a different subsequent evolution, with a likely transition from EuBL to GL and from GL to AT around 3-4 weeks after the SSW along with their contrasting large-scale weather impacts (Beerli and Grams, 2019). These findings corroborate that the presence of either EuBL or GL regime at SSW onset will allow us to disentangle the difference in the subsequent evolution, and hence to determine if and when a "downward impact" of the SSW is expected. This is highly relevant for subseasonal forecasting.

5. Several case studies of weather regimes at SSW onset confirm the above conclusions, e.g. 1987 (GL), 2001 (EuBL), and 2018 (cyclonic regimes). The 2019 SSW event, in contrast, occurred in an atypical manner during an AR regime, which likely resulted in the lack of a canonical response after the event.

While these findings are limited by the small sample size of available SSW events, the rigorous statistical testing for significance and robustness performed here suggests that the large case-to-case variability in the tropospheric response to SSWs is to a substantial extent owed to the state of the tropospheric circulation at the onset of the SSW. Our findings confirm that while the stratosphere does not represent the sole forcing of the tropospheric state, for many events it may be able to nudge the tropospheric flow into a particular direction by suppressing some weather regimes and by favoring others. We here show that the susceptibility of the troposphere to the stratospheric nudging depends strongly on the tropospheric state at the time of the SSW. While currently, prediction systems are often unable to forecast if a surface response is to be expected at the time of occurrence of the SSW event, our findings suggests that the presence or absence – and in fact the timing – of a surface impact following SSW events might be predictable based on the weather regime at the onset of the SSW event. This could significantly improve the subseasonal prediction of tropospheric winter weather over Europe.

*Data availability.* The ERA-interim reanalysis data (Dee et al., 2011) is available from ECMWF at https://apps.ecmwf.int/datasets/data/interim-full-daily/.





**Appendix A**

*Author contributions.* The authors together initiated and designed the study and all authors contributed to data analysis, discussion of results, and writing.

*Competing interests.* The authors declare no competing interests.

*Acknowledgements.* Support from the Swiss National Science Foundation through project PP00P2_170523 to DD is gratefully acknowledged. The contribution of CMG was supported by the Helmholtz Association as part of the Young Investigator Group "SPREADOUT" (grant VH-NG-1243). Data analysis and visualisation were performed using the NCAR Command Language (UCAR/NCAR/CISL/VETS, 2014).



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



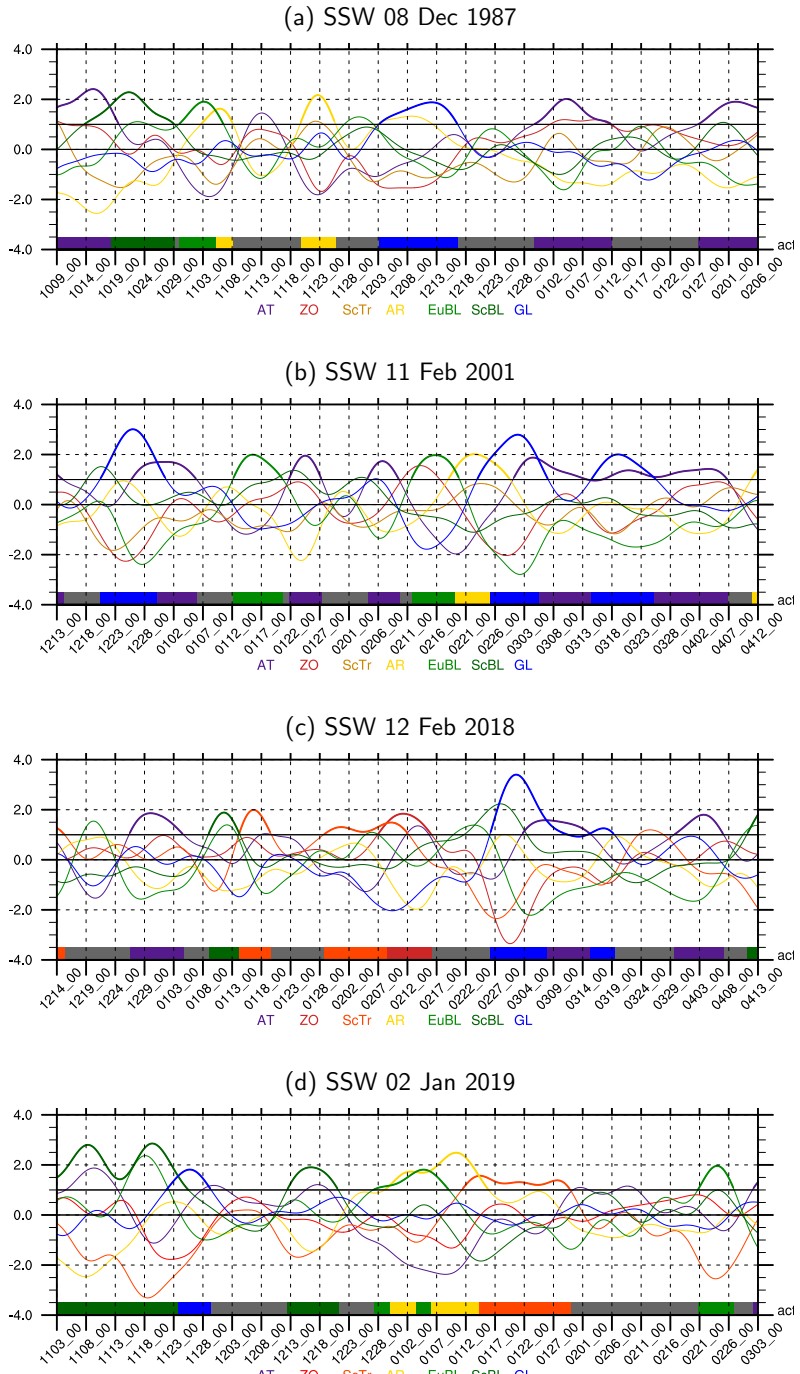

**Figure 6.** Weather regime index for each of the seven regimes (coloured lines in units of standard deviation of the absolute projections of normalized geopotential height onto the cluster mean, see Michel and Rivière (2011)) for ±60days centred around the SSW central date for the SSW case studies in winter (a) 1987/88, (b) 2000/01, (c) 2017/18, and (d) 2018/19 computed as the normalized projection of the instantaneous 500 hPa geopotential height anomaly on the seven cluster mean anomalies (cf. Section 2 and Grams et al. (2017)).





**Figure A1.** Composite mean 10-day low-pass filtered 500 hPa geopotential height anomaly (shading, every 20 gpm), and mean absolute 500 hPa geopotential height (black contours, every 20 gpm) for all winter days in ERA-Interim (DJF, 1979-2015) attributed to one of the 7 weather regimes (a-g) and the climatological mean (h). Regime name and relative frequency (in percent) are indicated in the sub-figure captions.



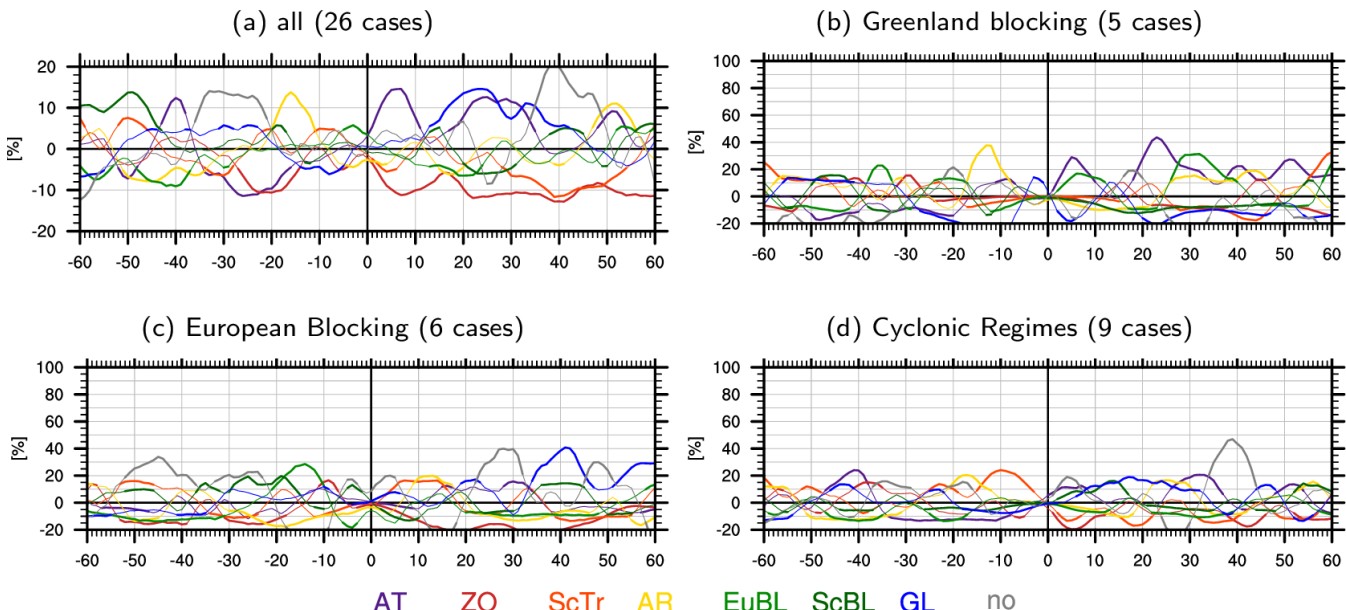

**Figure A2.** As Figure 2 but for the *anomalous* frequency of weather regimes for a 5-day period centred on the onset of the SSW event (lag 0) relative to the mean of the climatological distribution derived from 1000 Monte-Carlo samples: (a) all SSW events, (b) Greenland blocking, (c) European blocking, and (d) cyclonic regimes. Note the different y-axis in (a). Bold parts of the lines indicate significant deviations from climatology (see text for details). Note that anomalous frequencies at lag 0 in c, d are - by construction - close to zero as the same regime is prescribed for computing the mean from the 1000 Monte-Carlo samples.

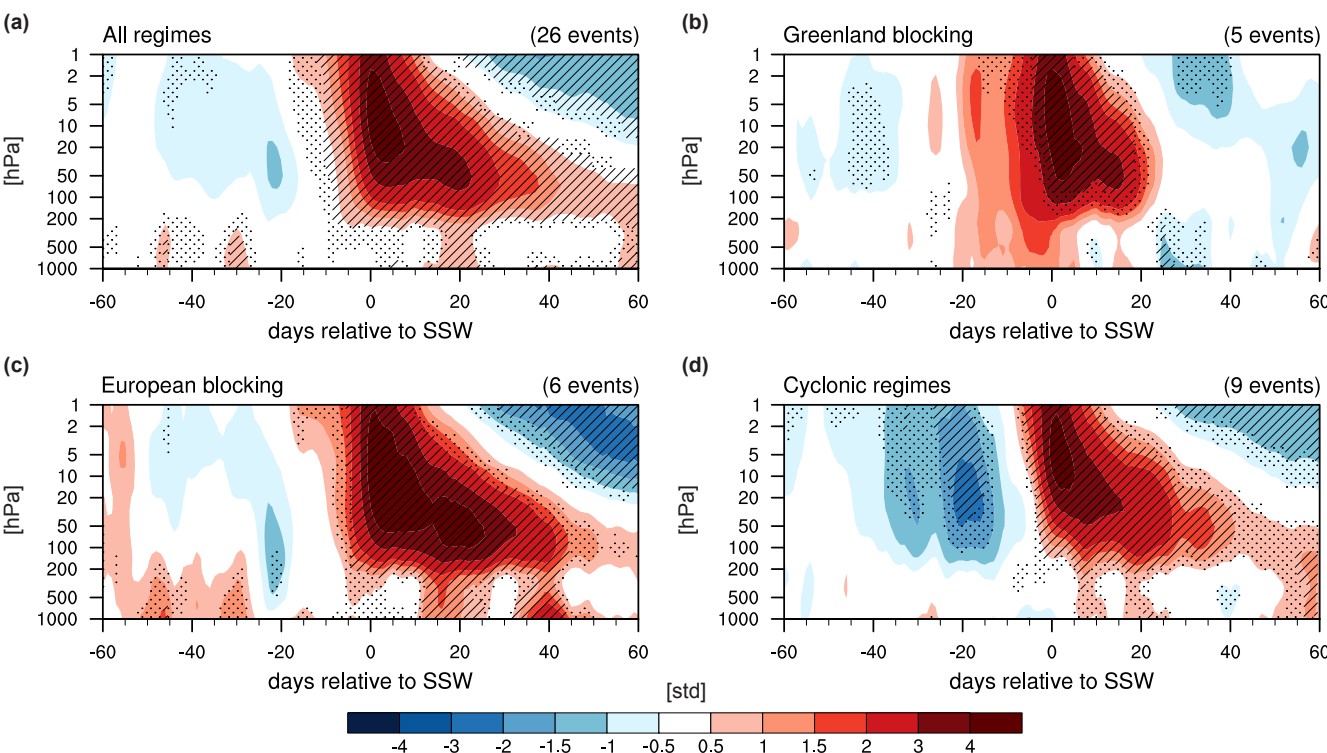

**Figure A3.** As Figure 3 but for the full longitude range, i.e. for the polar cap poleward of 60°N.





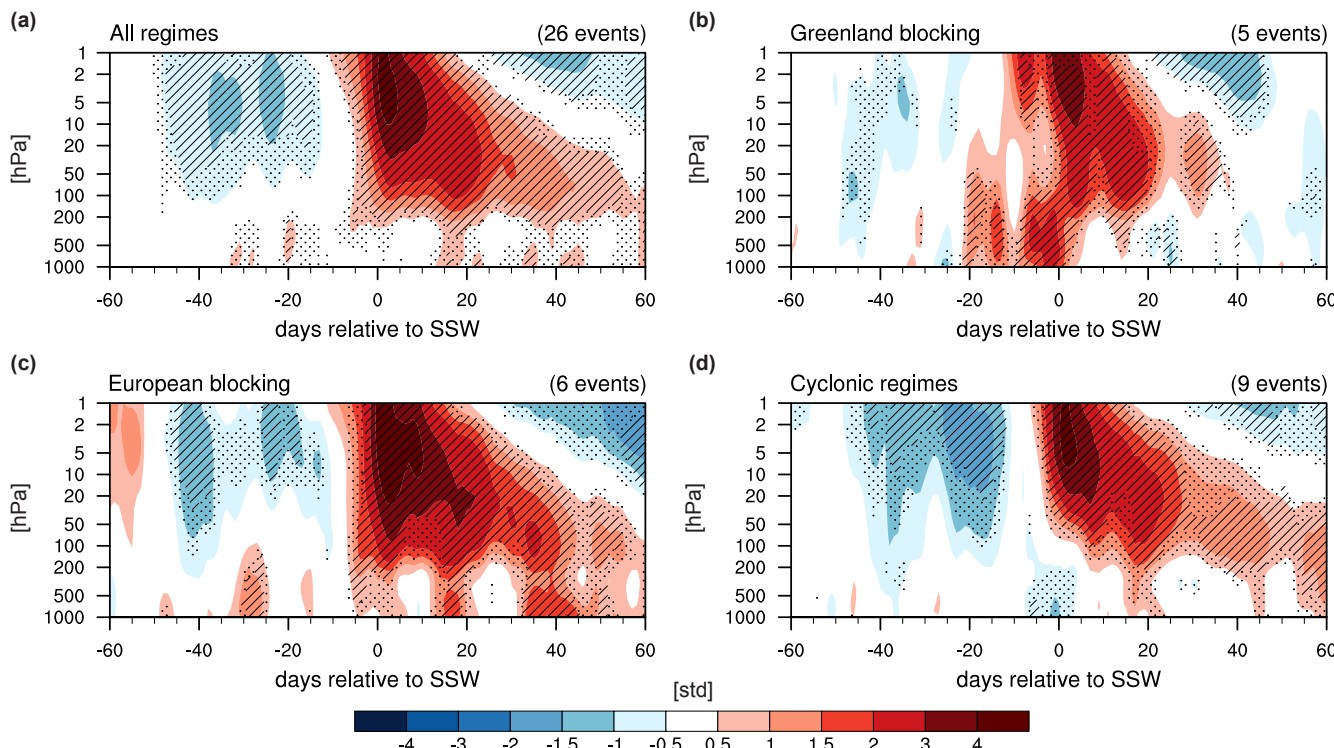

**Figure A4.** Standardized geopotential height anomalies for the sector -80°E to 40°E / 60°N to 90°N for (a) all SSW events, and (b-d) sub-divided by the weather regime that is dominant at the onset of the SSW as indicated by the titles of the panels. Robustness is assessed using confidence intervals by resampling the SSW events 100 times with repetition. If the magnitude of the anomaly exceeds the interquartile or the 10th-90th percentile ranges, the anomaly is considered robust (stippling) or highly robust (hatching). See text for details.

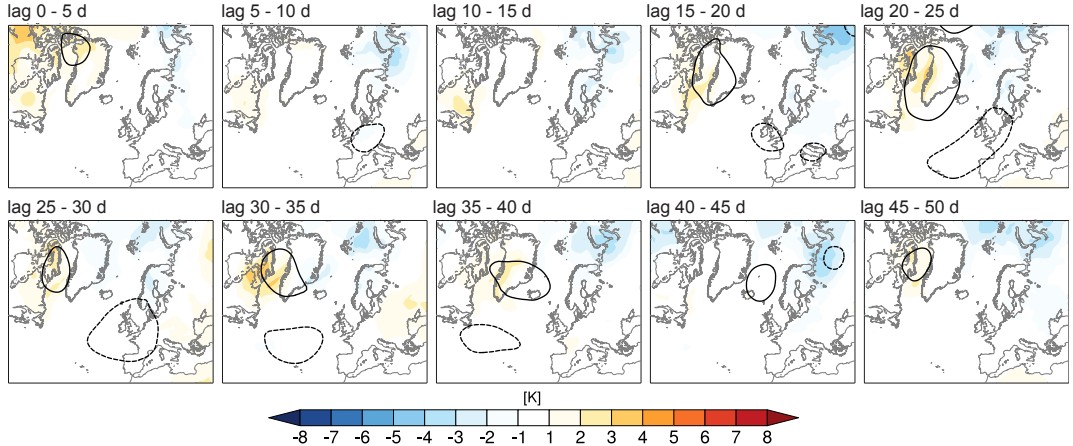

**Figure A5.** As figure 5 but averaged over all SSWs.



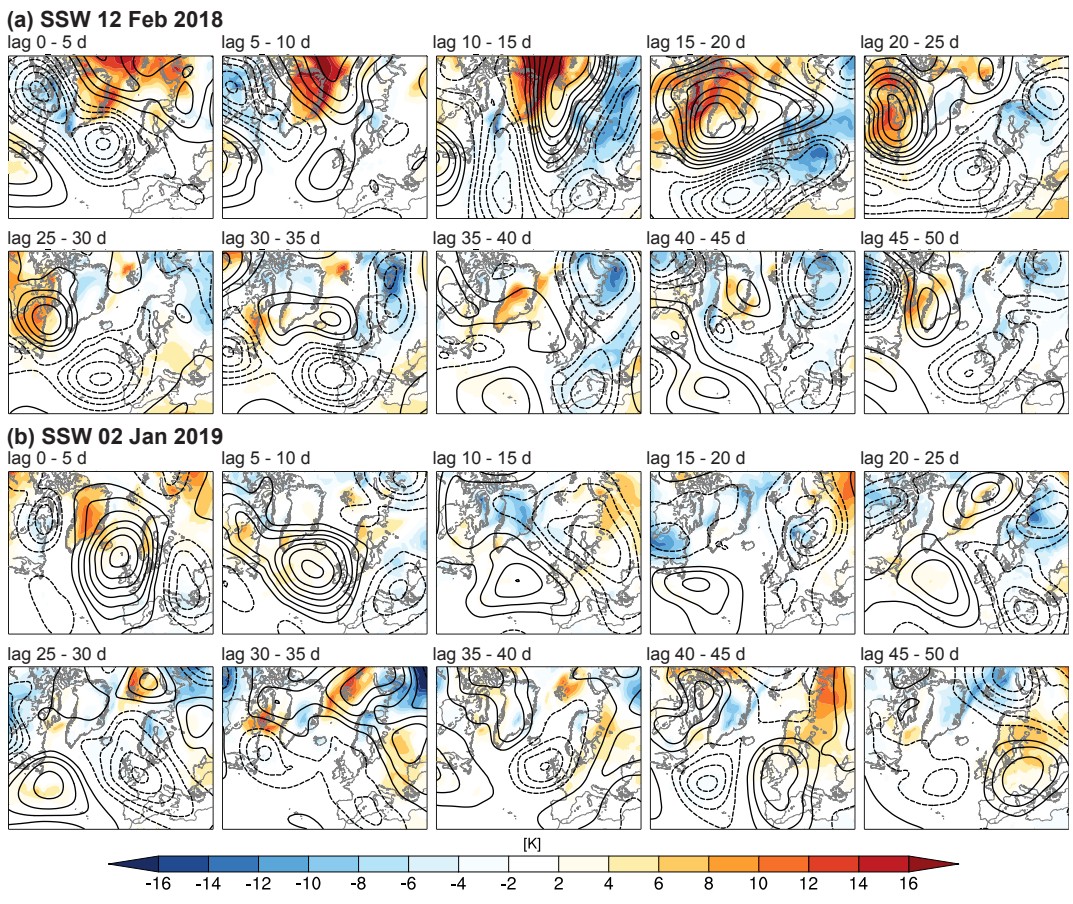

**Figure A6.** As figure 5 but for the SSWs on (a) 12 Feb 2018 and (b) 02 Jan 2019. Note the different scale for the temperature anomaly.