# Peer review of "The role of North Atlantic-European weather regimes in the surface impact of sudden stratospheric warming events"

_Weather and Climate Dynamics, 2019_

## Referee Comment (RC1) · Anonymous Referee #1 · 11 Feb 2020

General comments: The paper investigates response in the troposphere in the North Atlantic/European (NAE) region following stratospheric sudden warming (SSW) events in reanalysis data. The study finds that Greenland blocking and Atlantic trough (AT) are more likely weather regimes weeks after SSWs. In addition, the study investigates the role of tropospheric weather patterns during SSW onset in the subsequent tropospheric response. It is found that it is mostly for SSWs with European blocking at their onset that the canonical response of cold surface extremes over Europe is observed weeks following SSWs. In contrast, for SSWs with AT at the onset, mild conditions over NAE region after SSWs are observed. The remaining tropospheric flow patterns at the onset of SSWs were not associated with clear surface response following SSWs.

[Figure]

Given the large case to case variability of surface responses following SSWs, this study is a step in the right direction in trying to further understand when an SSW is likely to be followed by surface extremes. The paper shows that not all SSWs are followed by anomalous tropospheric weather patterns, therefore suggesting that caution must be exercised when generalising results from composite analysis involving all SSWs. Such knowledge is important for subseasonal to seasonal predictability when trying to assess if the downward impact from SSW is to be expected.

I commend the authors on statistical rigour and strongly encourage to carry out similar analysis (in the future) in the context of sub seasonal to seasonal prediction models where the robustness of the results to sampling uncertainty and the impact on predictability can both be assessed. I recommend this paper for publications and have only very minor comment detailed below.

Specific comment:

It would be helpful if the authors could put their study in the context of previously published studies that assessed which SSWs give stronger response. In particular the persistence and amplitude of lower stratospheric anomalies following SSWs are known to affect surface response (Hitchcock et al, 2013, JClim; Kodera et al., 2016, JGR; Runde et al., 2016, GRL; Karpechko et al, 2017; QJRMS; Polichtchouk et al, 2018, JAS). For example, is there any evidence to suggest that SSWs that had European Blocking at the onset also have larger and longer lasting anomalies in the lower stratosphere than the other cases?

Technical comment:

P7, L180: "NAM), the number"->"NAM), and the number"
* * *

---

## Referee Comment (RC2) · Anonymous Referee #2 · 17 Feb 2020

Summary

The study asks whether the North Atlantic European (NAE) weather regime present at the onset of an SSW has a bearing on the subsequent evolution of the tropospheric state. The topic fits within the scope of WCD and a study on this topic fits more broadly into a body of literature that has investigated possible factors to explain why some SSWs appear to couple to the troposphere and others do not.

The study uses ERA-Interim reanalysis data (26 SSWs) and views these through the lens of 7 NAE weather regime types introduced in earlier work by one of the co-authors. The authors conclude that European Blocking at the time of SSW onset favours Green-

land Blocking in the subsequent weeks, while Greenland blocking at onset favours an Atlantic Trough following the SSW. One major limitation of the study is the small sample of observed SSWs, which are then subdivided across the 7 regimes. This leaves only small samples for each subcategory. Earlier studies provide a cautionary tale about interpreting small subsets of SSWs (e.g., Mitchell et al. (2013) and Maycock and Hitchcock (2015)), and the authors fall foul to some of these issues.

The authors undertake bootstrap analyses to test for the significance of results, but this is mainly comparing to samples drawn from non-SSW periods. If the purpose is to test whether knowledge of the NAE state at the onset of an SSW can provide additional knowledge over and above knowledge of an SSW, the null hypothesis should be either that the tropospheric state following SSWs with a given day-0 regime is not distinguishable from that for all SSWs and/or that it is not distinguishable from SSWs with a different day-0 regime. This requires calculating differences (and their significance) between the regime subsets.

The authors also make no attempt to rule out other confounding factors that might affect their interpretation of the role of NAE regimes. For example, studies have found a relationship between the amplitude of lower stratospheric anomalies around the onset and the subsequent tropospheric NAM response. This was also pointed out by reviewer 1, but I think it is hugely important for the interpretation of the present results. The manner of presentation implicitly assumes the differences are a consequence of the day-0 regime, but since no other factors are tested for or displayed it is impossible to determine whether this is the case. This is especially pertinent given the small sample sizes being dealt with.

Overall, while the topic itself is potentially interesting, I found the manuscript disappointing both in terms of setting out the motivation for why/how the NAE state could have a long-lasting impact on the subsequent response and in terms of weaknesses in the analysis that I did not feel support the conclusions for the added value of knowing an SSW has occurred AND the day-0 NAE regime as compared to simply knowing an

SSW has occurred. I therefore recommend to reject the manuscript in its current form.

Recommendation: Reject.

Major comments

1) Hypotheses and statistical tests.

a) Your statistical test in Fig. 2 and 3 asks whether the SSW periods are different from non-SSW periods (climatology). This is fine for Figs 2a and 3a, where you ask about the overall signal of SSWs compared to no-SSWs, but what you are asking in Fig 2b-d is whether knowledge of the day-0 NAE regime provides extra information over and above the general knowledge of an SSW. Step 1) is there an SSW? Step 2) if yes, what is the NAE regime? Therefore, to my mind the relevant test is whether panels (b-d) are different from each other and/or different from (a). The same applies to Fig. 3. See e.g., important lessons from a parallel case on whether split vs. displacement SSWs show different coupling. Mitchell et al. (2013) performed similar analysis to that here for the NAM, but instead stratifying events based on split and displacement types (rather than on NAE type); importantly they neglected to test the significance of their differences, which was later done by Maycock and Hitchcock (2015) who estimated that the difference is not significant. You could do something similar here constructing a bootstrap distribution of the difference between two sets of N SSW samples.

b) Fig. 3: These dripping paint diagrams are notoriously sensitive to sampling uncertainty and for such small sample sizes I strongly question their representativity. Charlton and Polvani (2007) stated in relation to their assessment of the impacts of split and displacements (p.462, Section 6) "We started our analysis by first constructing time–height composites of the NAM index for the two types of SSW. However, the structure of the NAM index for the two types of SSW was found to be extremely sensitive, particularly in the troposphere: the size and timing of the composite NAM index anomalies following the events could be substantially altered by adding or removing even a single event. Hence, composite time–height NAM plots could not be used to examine differences in tropospheric impact between the vortex splits and vortex displacements." They made this point in relation to splits and displacements which have bigger sample sizes than those considered here. A similar point was also made by Maycock and Hitchcock (see e.g., their Fig. 3). Charlton and Polvani (2007) instead use the integrated NAM index to assess differences between splits and displacements. You could try an approach along these lines instead.

c) No attempt is made to rule out other possible associations than the NAE regime at day-0. For example, what if there is an indirect relationship to some other factor, such as the amplitude and persistence of the stratospheric anomalies themselves (e.g., Karpechko et al., 2017). There is some hint in Fig. 3 that the character of the stratospheric anomalies is different for these particular subsets of events; might that not be important? The sample sizes available here are very limiting in being able to say what is going on. To my mind, other more effective studies on related topics of downward coupling have combined reanalysis and model results (Karpechko et al., 2017; Maycock and Hitchcock, 2015). Reviewer 1 talks about following this up with a study on S2S models. If the authors do plan this, my recommendation would be to combine the current results with such a model study.

d) L193-196 "The immediate positive geopotential height anomalies and the weak tropospheric response in the aftermath of the event are archetypal for SSWs with GL at the onset and not the result of cancellations in the composites. Indeed, they are also evident for individual SSWs, such as the SSW on 8-Dec-1987 that exhibited a dominant GL regime for an extended period around the SSW onset (Figs. 4a and 6a)." I find Figures 4 and 6 completely uninformative. You have chosen examples to support your proposed hypotheses, but the key information is what comes from the behaviour across all events, as shown in Fig 2 and 3. To give just one example, you could have chosen instead the GL event on 9 Feb 2010 which shows the GL regime for 3 weeks after the onset. Presumably this event is not "archetypal" but it is one of your 5 cases. I would also argue you cannot conclude something is "archetypal" when

you have only 5 events. I suggest removing these arguably cherry picked case studies and providing more comprehensive evidence for a detectible difference between the subsets discussed would improve the manuscript. This also applies to the discussion of the 2018/2019 events, which I found too cursory and descriptive to provide any real insight.

2) Existence of plausible mechanism(s). L53-57: "Given the large tropospheric internal variability and the influence of other remote effects mentioned above, it appears plausible that also the tropospheric state at the time of occurrence of an SSW plays an important role in shaping the characteristics of its downward impact. For example, tropospheric jet characteristics have been suggested to affect the downward impact of SSW events (Chan and Plumb, 2009; Garfinkel et al., 2013)."

I appreciate the goal of the study is not to explain but rather to diagnose, but this point is central to the whole premise of the study. However, the studies cited here are highly idealised and explore a much wider range of basic states than is plausible for the real world in idealised models that do not produce the type of NAE regime behaviour described here. I therefore do not agree this is supporting evidence for the proposed hypothesis. Indeed, no mechanism or theory is provided to justify why, or in what way, the tropospheric NAE state at day-0 would influence the subsequent NAE state up to +60 days. If that is the motivation to pursue this analysis, then some hypothesis for a mechanism is needed to explain an effect that extends far beyond the characteristic decorrelation timescale of the NAE circulation. It appears the proposal is for a vague mechanism related to internal tropospheric dynamics. However, this seems to defy the premise of why SSWs are useful for predictability in the first place, which is that their intrinsic timescale is much longer than the 'memory' of the tropospheric circulation. To make a more convincing case for this, more discussion is needed around the persistence characteristics of the regimes themselves and the canonical transitions amongst the regimes to put the behaviour following SSWs into context.

3) Timescales. Related to 2), a more careful description of the relevant timescales is

needed in the introduction. Since the downward influence of SSWs may last for up to 6-8 weeks are the authors proposing that the NAE regime on day 0 bears some relevance for the response in week 6? Or are the authors talking about the downward coupling over a shorter period following the onset, e.g. in week 1? This does not become clear until one gets into the results, so some explicit statements on timescales in the abstract and introduction would clarify this and this should tie into the discussion of mechanisms.

4) Dataset. Why is ERA-Interim used and not a longer reanalysis like JRA-55 which contains more SSWs (41 compared to 26 in Butler et al (2017))? For rare events, the benefits of increased sample size can outweigh other uncertainties in the pre-satellite era (Hitchcock, 2019).

Hitchcock, P.: On the value of reanalyses prior to 1979 for dynamical studies of stratosphere–troposphere coupling, Atmos. Chem. Phys., 19, 2749–2764, https://doi.org/10.5194/acp-19-2749-2019, 2019.

5) Other studies. The introduction ignores important information on past efforts (and their degree of success) in identifying stratospheric factors that may influence downward coupling, e.g.:

Charlton, A.J. and L.M. Polvani, 2007: A New Look at Stratospheric Sudden Warmings. Part I: Climatology and Modeling Benchmarks. J. Climate, 20, 449–469,https://doi.org/10.1175/JCLI3996.1

Karpechko, A.Y., Hitchcock, P., Peters, D.H.W. and Schneidereit, A. (2017), Predictability of downward propagation of major sudden stratospheric warmings. Q.J.R. Meteorol. Soc., 143: 1459-1470. doi:10.1002/qj.3017

Mitchell, D.M., L.J. Gray, J. Anstey, M.P. Baldwin, and A.J. Charlton-Perez, 2013: The Influence of Stratospheric Vortex Displacements and Splits on Surface Climate. J. Climate, 26, 2668–2682, https://doi.org/10.1175/JCLI-D-12-00030.1

Maycock, A. C., and Hitchcock, P. (2015), Do split and displacement sudden stratospheric warmings have different annular mode signatures?, Geophys. Res. Lett., 42, 10,943– 10,951, doi:10.1002/2015GL066754.

Nakagawa, K. I., and Yamazaki, K. ( 2006), What kind of stratospheric sudden warming propagates to the troposphere? Geophys. Res. Lett., 33, L04801, doi:10.1029/2005GL024784.

Runde, T., Dameris, M., Garny, H., and Kinnison, D. E. ( 2016), Classification of stratospheric extreme events according to their downward propagation to the troposphere, Geophys. Res. Lett., 43, 6665– 6672, doi:10.1002/2016GL069569.

Seviour, W. J. M., Gray, L. J., and Mitchell, D. M. (2016), Stratospheric polar vortex splits and displacements in the high‐top CMIP5 climate models, J. Geophys. Res. Atmos., 121, 1400– 1413, doi:10.1002/2015JD024178.

Specific comments

L6: following the weeks after an SSW –> in the weeks following an SSW

L45-49 you need to add e.g., to these reference lists as they are highly selective

L54 remove 'also'

L55 occurrence of an SSW also plays

L65-67 "Prior work based on this extended regime definition revealed important differences in the surface weather response to the state of the stratosphere, which remain hidden using the canonical four NAE regimes (Papritz and Grams, 2018; Beerli and Grams, 2019)." It seems this needs expanding as this is important justification for the current approach of using seven regimes rather than four. What specifically is missed? Also what did Papritz and Grams, 2018 and Beerli and Grams, 2019 show in relation to the two questions investigated here? Did they analyse similar things? What did they find?

L113-115 These statements are not visible from Figure 1 without some specific information on frequencies given in the text or in a table. Also in Figure 2a I see a peak in AR between -20 to -10 days but I cannot see a clear higher frequency for ScTr compared to all the other states. Rather the peak in EuBL immediately before onset seems to be a clearer feature for "all events".

L138-139: All the subsets you are dealing with are small sample sizes. The no regime cases can provide a useful null hypothesis, i.e. what anomalous regime frequencies can apparently arise without a specific regime being identified at day 0?

Fig. 2 and A2: The choice to use 5-day running means and to test for significance on that basis has implications related to the intrinsic persistence characteristics of each regime. But these timescales differ – e.g., from Fig. 1 it appears the canonical persistence timescale of GL is longer than, say, EuBL. It needs to be mentioned how the authors have accounted for the intrinsic persistence of each regime in choosing the smoothing window. Also, do you need to account for autocorrelation in your statistical tests?

I find Figure A2 more informative than Figure 2 since what you wish to highlight is the anomalous frequencies associated with particular subsets of data not the absolute frequencies. This is more clearly seen in Figure A2. For example, it becomes clear that the significant anomalies in AT at lags -35 to -15 is because the frequency is anomalously low (i.e. a negative anomaly). I suggest switching them and putting Fig. A2 in the main text and Fig. 2 in the Appendix.

L163-164 "Given the strong influence of the tropospheric state at the time of the onset of an SSW on the weather regime frequencies in the subsequent days" I don't agree you have demonstrated this in Section 3. See major comment 1.

L170-171 These are weaker thresholds than one would typically associate with "robust" and "highly robust"

L 179-187 and Fig. A3: Why are the tropospheric Z anomalies so weak? Is it a matter of plotting (e.g., contour intervals)? The SSW compendium NAM composite for the same set of events in ERA-Interim (see Fig. 1 below) looks quite different from your Z anomalies (Butler et al., 2017).

Fig 5: Is any statistical testing applied to the anomalies? The caption does not mention it.

Typographical

Figures – the two shades of green for EuBL and ScBL are hard to differentiate

––––––––––––––––––––––

[Figure]

[Figure]

**Fig. 1.** SSW compendium composite NAM anomaly for SSWs in ERA-Interim (Butler et al., 2017).

---

## Author Comment (AC1) · 2 Apr 2020

**Response to Reviewers**

We would like to thank both reviewers for their time and effort to review our manuscript. We are happy that Reviewer 1 acknowledges our effort in developing statistical tests to detect significant and robust signals despite the sparse sampling size. We thank Reviewer 2 for the constructive criticism. The reviewer comments will help to balance the discussion of the various aspects related to stratosphere-troposphere coupling around SSW events and therefore to provide a contribution by adding a tropospheric viewpoint to this longstanding discussion.

Our intention was and is to use tropospheric weather regimes to stress the variety of the tropospheric response, and in particular that the tropospheric response can be divided into major categories that also tend to influence the subsequent tropospheric evolution. Our intention is also to raise even more awareness that the canonical response of NAO negative conditions and a cold surge in Europe, as indicated from composite studies, can often be misleading. We concur that due to our wording the initial version implies too much causality. We also concur that if we had intended to show causality we would have to discuss the underlying mechanisms explaining why the tropospheric state of the NAE sector affects the aftermath of an SSW in the troposphere. Instead, with the study at hand we aim to shed light on the potential role of the troposphere in the response to stratospheric variability and to suggest weather regimes at SSW onset as a potential indicator of the subsequent tropospheric impact. This does not exclude other processes but might be an important additional factor.

In order to make this intention of our study clearer and to achieve a more balanced discussion of our results, we will rewrite our study and thereby accommodate the reviewer comments. In a revised manuscript we will highlight the potential role of the troposphere in stratosphere-troposphere coupling following SSW events. In particular, we will more carefully introduce the tropospheric state in the NAE sector in terms of the refined 7 weather regimes as a potential factor which preconditions the impact of SSWs.

**Answer to major reviewer comments**

In the following we comment on the major points raised by both reviewers and how we aim to address these. This response will be followed by a detailed response to all reviewer comments and a revised manuscript.

1. **Extending the analysis to S2S prediction models:** We fully agree that extending the analysis to S2S prediction models will be extremely worthwhile. However, it is currently unknown to what extent complex prediction models are indeed able to represent the variety of tropospheric responses to stratospheric forcing. Although simplified models indeed show a role of the troposphere in the downward impact, as reviewer 2 points out, this has not been sufficiently tested in more complex models beyond the canonical response and selected case studies. From a preliminary

analysis of S2S prediction model data we anticipate large biases and a very complex role of the representation of stratosphere - troposphere coupling in prediction models. This will be complex to disentangle, and we will therefore not be able to cover the analysis of the model data in this study. We will comment on this in the revised version of the manuscript.

2. **Role of the lower stratospheric persistence in the downward response:** We agree that the persistence of lower stratospheric anomalies is a very important issue, and we will include a more thorough discussion of this matter and the relevant citations in the manuscript. An inital analysis revealed that 2 out of 5 SSW events with the GL regime at the onset have a persistent lower stratospheric temperature response, while 4 out of 6 EuBl cases have a persistent lower stratospheric temperature response. This goes in the right direction by indicating that the shorter (longer) persistence in the lower stratosphere for the SSWs associated with GL (EuBl) may add some support to the persistence of the tropospheric response, but the statistics are too small to provide a clear result. We will expand on this in more detail in the new manuscript.

3. **Statistical testing:** We fully agree that the number of SSWs in the observational record of the satellite era is very small. Thus, we have put significant effort into the design of the statistical tests, as also commended by Reviewer 1, to take sampling uncertainty into account and obtain meaningful results. We recognize that in the manuscript this procedure may not have been explained in sufficient detail. We will expand on this in the revised version of the manuscript. Furthermore, we clarify the procedure again in the following.

The overarching question we address in this study is whether after SSWs we can detect robust tropospheric geopotential height anomalies and whether these anomalies are significantly different from situations without an SSW. Hence, the relevant null hypothesis is that the tropospheric geopotential height anomalies after SSWs are indistinguishable from geopotential height anomalies commonly occurring in the absence of an SSW. The testing procedure follows two important steps:

1. First, we assess the **robustness** of the samples by performing a Monte Carlo resampling. For that purpose, the dripping paint plots are re-computed by resampling the original samples 100 times with repetitions. This yields confidence intervals of the dripping paint plots, estimating the uncertainty inherent in each sample. Due to the small sample size, these confidence intervals are relatively large.

2. Second, we compute 1000 random samples of the same size as the original sample but for random periods with the same weather regime(s) at the central date but no SSW occurring within $\pm 60$ days, yielding estimates of the distributions of geopotential height anomalies occurring in the absence of SSWs. Testing for **significance** is done by comparing the confidence intervals and distributions obtained from the random samples.

Following this procedure, we thus show that the anomalies in the ALL composite are not highly robust (Fig. A4a in the manuscript), indicative of the large variability in the tropospheric response.

Yet, the anomalies observed between 10-20 days after the SSW are statistically significantly different from non-SSW periods. Now, we can ask the question whether there are subsamples of all SSWs that show a more robust response that is also statistically different from periods with no SSW. We select these subsamples according to the weather regime present at the time of the SSW. Note that the null hypothesis here is still the same as for the ALL sample, namely that no significantly different anomalies occur. Thus, we find that particularly robust and significant anomalies occur if the SSW is timed with European blocking, for example.

[Figure]

Figure 1: Standardized geopotential height anomalies as in Fig. 3 in the manuscript for (a, b) Greenland blocking cases, (c, d) European blocking cases, and (e, f) cyclonic regimes. Stippling and hatching indicates statistically significantly different anomalies from the other two samples (as indicated in the title of each panel) obtained from an overlap of the confidence intervals by less than 25 % and 10 %, respectively.

The reviewer suggests to test for the null hypothesis that the geopotential height anomalies between samples are identical. This null hypothesis may be appropriate if we were interested in the question whether geopotential height anomalies between the samples are different from each other. While this may be an interesting alternate path of investigation, it is not the question we pose here. For the reviewer's interest, we have nevertheless done mutual tests of the Greenland blocking, European Blocking, and the cyclonic regimes samples (Figure 1). This reveals that the geopotential height anomalies in the European blocking case, for example, are significantly different from the Greenland blocking case. The difference is less significant with respect to the cyclonic

regimes samples (especially the anomalies before day 20).

We would like to stress that mutually testing the individual samples for difference does not a priori tell anything about whether the flow evolution in the presence of an SSW is different from that in the absence of an SSW. Similar significant / insignificant differences between samples may in principle also arise from samples obtained from random days with a given weather regime irrespective of whether an SSW occurred or not.

We hope that with these additional clarifications, we are able to convince the reviewer that our testing procedure is appropriate for the questions addressed in the study. In the revised manuscript, we will more carefully explain the statistical testing procedure in subsection 2.2 and discuss the appropriate null hypothesis.

Further, we do not agree that the no regime category would provide a suitable null hypothesis for the hypotheses addressed in this study. First, the no regime category arises due to a weak projection of the geopotential height anomalies in the Atlantic European sector on one particular cluster centroid. This does not necessarily imply that the geopotential height signal is particularly weak. Instead, this may just reflect a progression between regimes. Second, we aim to test whether the regime succession is different in the presence of an SSW compared to the case where no SSW took place. Thus, the null hypothesis is that the regime progression in the aftermath of an SSW is the same as in cases with the same initial regime but no SSW.

[Figure]

Figure 2: As Figure 4 in the manuscript but for the SSW on Feb 9, 2010.

4. **Case studies:** As suggested by the reviewer we have here included the dripping paint plot for the 2010 SSW event (Figure 2). The event again supports the general structure of the events that

are dominated by GL at the onset of the SSW, with a strong tropospheric signal already before the onset, and the tropospheric response generally limited to the weeks just after the SSW event and including an AT response. We agree that the event exemplifies variability between events in the same sub-category, but this event as well as all others in the GL category indeed show a very different response to e.g. the EuBl cases. We will be more careful in presenting the case studies (if any) and reformulate this section.

5. **Persistence of the tropospheric weather regimes:** We will clarify the relevant timescales and persistence of the regimes in the new version of the manuscript. In particular, we are indeed not suggesting that the response 6-8 weeks after the SSW onset is dominated by the weather regime at lag 0. We will formulate this more carefully in the revised manuscript.

6. **Use of pre-satellite date:** Thank you for this comment. We had considered including pre-satellite data for the original manuscript, but we were worried that the atmospheric state would be much less constrained and rely more on the model itself, hence we decided against using additional SSWs with a poorer representation of the atmospheric circulation.

7. **References / technical comments:** We will of course be happy to discuss the additional references suggested by the reviewers in our revised manuscript, as well as the minor and technical comments by both reviewers.

---

## Author Response (AR1)

**Response to Reviewers**

We would like to thank both reviewers for their time and effort to review our manuscript. We here list the major changes that we made in the manuscript during the revision:

- Role of the lower stratospheric persistence in the downward response: We have now included an extended discussion and analysis of the lower stratospheric persistence in the Introduction, Results, and Discussion sections of our manuscript.
- Role of weather regimes: We have rewritten the manuscript to omit causal statements and emphasise more that regimes at the onset might be an indicator for the potential surface impact of an SSW.
- Statistical testing: We have carefully re-written and significantly extended the section about statistical testing. In particular, the hypotheses have now been clearly formulated.
- Case studies: We have removed the case studies at the suggestion of the reviewer.
- Persistence of the tropospheric weather regimes and causality: We have clarified the relevant timescales and persistence of the regimes in the new version of the manuscript. We have also clarified the definition of the weather regimes and their evolution. In particular, we are indeed not suggesting that the response 6-8 weeks after the SSW onset is a direct cause of the weather regime at lag 0.
- We have limited the Figure showing the surface temperature response to lags 0 25 days allowing us to include the composite for all SSWs for better comparability.

Please find below the detailed responses (in *cursive*) to the reviewers' comments and suggestions. All line indications refer to the new (annotated) manuscript at the end of the reviewer response.

**Reviewer 1:**

General comments: The paper investigates response in the troposphere in the North Atlantic/European (NAE) region following stratospheric sudden warming (SSW) events in reanalysis data. The study finds that Greenland blocking and Atlantic trough (AT) are more likely weather regimes weeks after SSWs. In addition, the study investigates the role of tropospheric weather patterns during SSW onset in the subsequent tropospheric response. It is found that it is mostly for SSWs with European blocking at their onset that the canonical response of cold surface extremes over Europe is observed weeks following SSWs. In contrast, for SSWs with AT at the onset, mild conditions over NAE region after SSWs are observed. The remaining tropospheric flow patterns at the onset of SSWs were not associated with clear surface response following SSWs.

Given the large case to case variability of surface responses following SSWs, this study is a step in the right direction in trying to further understand when an SSW is likely to be followed by surface extremes. The paper shows that not all SSWs are followed by anomalous tropospheric weather patterns, therefore suggesting that caution must be exercised when generalising results from composite analysis involving all SSWs. Such knowledge is important for subseasonal to seasonal predictability when trying to assess if the downward impact from SSW is to be expected.

I commend the authors on statistical rigour and strongly encourage to carry out similar analysis (in the future) in the context of sub seasonal to seasonal prediction models where the robustness of the results to sampling uncertainty and the impact on predictability can both be assessed. I recommend this paper for publications and have only very minor comment detailed below.

We would like to thank the reviewer for their positive evaluation of our manuscript. We fully agree that extending the analysis to S2S prediction models will be very worthwhile, and we have commented on this in the new manuscript at the end of the Discussion section (lines 311 - 319).

**Specific comment:**

It would be helpful if the authors could put their study in the context of previously published studies that assessed which SSWs give stronger response. In particular the persistence and amplitude of lower stratospheric anomalies following SSWs are known to affect surface response (Hitchcock et al, 2013, JClim; Kodera et al., 2016, JGR; Runde et al., 2016, GRL; Karpechko et al, 2017; QJRMS; Polichtchouk et al, 2018, JAS). For example, is there any evidence to suggest that SSWs that had European Blocking at the onset also have larger and longer lasting anomalies in the lower stratosphere than the other cases?

We agree that the persistence of lower stratospheric anomalies is a very important issue, and we have included a more thorough discussion of this matter and the relevant citations in the manuscript. Our analysis reveals that 2 out of 5 SSW events with the GL regime at the onset have a persistent lower stratospheric temperature response, while 4 out of 6 EuBl cases have a persistent lower stratospheric temperature response. This goes in the right direction by indicating that the shorter (longer) persistence in the lower stratosphere for the SSWs associated with GL (EuBl) may add some support to the persistence of the tropospheric response, but the statistics are too small to provide a clear result. We have included a more extensive discussion of the above references and lower stratospheric persistence into the manuscript (Introduction: lines 44 - 55, Results: lines 232-245, Discussion: lines 303 - 310). For Polichtchouk et al 2018, JAS: After discussion with Inna Polichtchouk we decided that her 2018 GRL paper might be most relevant here, hence we have now included a reference to her 2018 GRL paper instead.

**Technical comment:**

P7, L180: "NAM), the number"  $\rightarrow$  "NAM), and the number"

Thanks for spotting this, this has been corrected (line 202).

**Reviewer 2**

**Summary**

The study asks whether the North Atlantic European (NAE) weather regime present at the onset of an SSW has a bearing on the subsequent evolution of the tropospheric state. The topic fits within the scope of WCD and a study on this topic fits more broadly into a body of literature that has investigated possible factors to explain why some SSWs appear to couple to the troposphere and others do not. The study uses ERA-Interim reanalysis data (26 SSWs) and views these through the lens of 7 NAE weather regime types introduced in earlier work by one of the co-authors. The authors conclude that European Blocking at the time of SSW onset favours Greenland Blocking in the subsequent weeks, while Greenland blocking at onset favours an Atlantic Trough following the SSW. One major limitation of the study is the small sample of observed SSWs, which are then subdivided across the 7 regimes. This leaves only small samples for each subcategory. Earlier studies provide a cautionary tale about interpreting small subsets of SSWs (e.g., Mitchell et al. (2013) and Maycock and Hitchcock (2015)), and the authors fall foul to some of these issues. The authors undertake bootstrap analyses to test for the significance of results, but this is mainly comparing to samples drawn from non-SSW periods. If the purpose is to test whether knowledge of the NAE state at the onset of an SSW can provide additional knowledge over and above knowledge of an SSW, the null hypothesis should be either that the tropospheric state following SSWs with a given day-0 regime is not distinguishable from that for all SSWs and/or that it is not distinguishable from SSWs with a different day-0 regime. This requires calculating differences (and their significance) between the regime subsets. The authors also make no attempt to rule out other confounding factors that might affect their interpretation of the role of NAE regimes. For example, studies have found a relationship between the amplitude of lower stratospheric anomalies around the onset and the subsequent tropospheric NAM response. This was also pointed out by reviewer 1, but I think it is hugely important for the interpretation of the present results. The manner of presentation implicitly assumes the differences are a consequence of the day-0 regime, but since no other factors are tested for or displayed it is impossible to determine whether this is the case. This is especially pertinent given the small sample sizes being dealt with. Overall, while the topic itself is potentially interesting, I found the manuscript disappointing both in terms of setting out the motivation for why/how the NAE state could have a long-lasting impact on the subsequent response and in terms of weaknesses in the analysis that I did not feel support the conclusions for the added value of knowing an SSW has occurred AND the day-0 NAE regime as compared to simply knowing an SSW has

occurred. I therefore recommend to reject the manuscript in its current form. Recommendation: Reject.

We thank the reviewer for the thorough evaluation of the manuscript. We will answer the specific comments below.

**Major comments**

1) Hypotheses and statistical tests.

a) Your statistical test in Fig. 2 and 3 asks whether the SSW periods are different from non-SSW periods (climatology). This is fine for Figs 2a and 3a, where you ask about the overall signal of SSWs compared to no-SSWs, but what you are asking in Fig 2b-d is whether knowledge of the day-0 NAE regime provides extra information over and above the general knowledge of an SSW. Step 1) is there an SSW? Step 2) if yes, what is the NAE regime? Therefore, to my mind the relevant test is whether panels (b-d) are different from each other and/or different from (a). The same applies to Fig. 3. See e.g., important lessons from a parallel case on whether split vs. displacement SSWs show different coupling. Mitchell et al. (2013) performed similar analysis to that here for the NAM, but instead stratifying events based on split and displacement types (rather than on NAE type); importantly they neglected to test the significance of their differences, which was later done by Maycock and Hitchcock (2015) who estimated that the difference is not significant. You could do something similar here constructing a bootstrap distribution of the difference between two sets of N SSW samples.

We fully agree that the number of SSWs in the observational record of the satellite era is very small. Thus, we have put significant effort into the design of the statistical tests, as also commended by Reviewer 1, to take sampling uncertainty into account and obtain meaningful results. We recognize that in the manuscript this procedure may not have been explained in sufficient detail. We have expanded on this in the revised version of the manuscript (section 2.2). Furthermore, we clarify the procedure again in the following.

The overarching question we address in this study is whether after SSWs we can detect robust tropospheric geopotential height anomalies and whether these anomalies are significantly different from situations without an SSW. Hence, the relevant null hypothesis is that the tropospheric geopotential height anomalies after SSWs are indistinguishable from geopotential height anomalies commonly occurring in the absence of an SSW. The testing procedure follows two important steps:

1. First, we assess the **robustness** of the samples by performing a Monte Carlo resampling. For that purpose, the dripping paint plots are re-computed by resampling the original samples 100 times with repetitions. This yields confidence intervals of the dripping paint plots, estimating the uncertainty inherent in each sample. Due to the small sample size, these confidence intervals are relatively large.  Second, we compute 1000 random samples of the same size as the original sample but for random periods with the same weather regime(s) at the central date but no SSW occurring within ±60 days, yielding estimates of the distributions of geopotential height anomalies occurring in the absence of SSWs. Testing for significance is done by comparing the confidence intervals and distributions obtained from the random samples.

Following this procedure, we thus show that the anomalies in the ALL composite are not robust at the 10% level but at the 25% (Fig. A4a in the manuscript), indicative of the large variability in the tropospheric response. Yet, the anomalies observed between 10-20 days after the SSW are statistically significantly different from non-SSW periods at the 10% (Fig. 3a in the manuscript). Now, we can ask the question whether there are subsamples of all SSWs that show a more robust response that is also statistically different from periods with no SSW. We select these subsamples according to the weather regime present at the time of the SSW. Note that the null hypothesis here is still the same as for the ALL sample, namely that no significantly different anomalies occur. Thus, we find that particularly robust and significant anomalies occur if the SSW is timed with European blocking, for example.

The reviewer suggests to test for the null hypothesis that the geopotential height anomalies between samples are identical. This null hypothesis may be appropriate if we were interested in the question whether geopotential height anomalies between the samples are different from each other. While this may be an interesting alternate path of investigation, it is not the question we pose here. For the reviewer's interest, we have nevertheless done mutual tests of the Greenland blocking, European Blocking, and the cyclonic regimes samples (Figure 1). This reveals that the geopotential height anomalies in the European blocking case, for example, are significantly different from the Greenland blocking case. The difference is less significant with respect to the cyclonic regimes samples (especially the anomalies before day 20).

We would like to stress that mutually testing the individual samples for difference does not a priori tell anything about whether the flow evolution in the presence of an SSW is different from that in the absence of an SSW. Similar differences between samples may in principle also arise from samples obtained from random days with a given weather regime irrespective of whether an SSW occurred or not.

We hope that with these additional clarifications, we are able to convince the reviewer that our testing procedure is appropriate for the questions addressed in the study. In the revised manuscript, we have more carefully explained the statistical testing procedure in subsection 2.2 and discuss the appropriate null hypothesis.

b) Fig. 3: These dripping paint diagrams are notoriously sensitive to sampling uncertainty and for such small sample sizes I strongly question their representativity. Charlton and Polvani (2007) stated in relation to their assessment of the impacts of split and displacements (p.462, Section 6) "We started our analysis by first constructing time - height composites of the NAM index for the

---

## Author Response (AR2)

**Response to Reviewers**

We would like to thank the reviewer for their time and effort to review our manuscript. The suggestions, in particular, to consider the dominant weather regime within a time interval instead of the regime at a fixed SSW onset time, are very valuable and helped to further improve the study. We here list the major changes that we made in the manuscript during the revision:

- We have re-written parts of the manuscript as suggested by the reviewer. In particular, the *Introduction* and *Discussion* sections have been adapted following the reviewer's suggestions.

- The *Methods* section has been adapted clarifying the SSW and WR definitions, and to clarify the regime definition and the 5-day persistence criterion we used.

- We have updated the regime definition, which now considers the dominant regime within $\pm 5$ days around the SSW central dates instead of the previous criterion limited to the day of SSW onset with a 5-day persistence criterion. All figures have been updated following this change, and the table has been replaced by a new version to account for these changes.

Please find below the detailed responses (in *cursive*) to the reviewer's comments and suggestions. All line indications refer to the new (annotated) manuscript at the end of the reviewer response.

**Reviewer 3:**

General comments: This is my first time reviewing the paper. The paper addresses (from a diagnostic standpoint) an important and relevant question in stratosphere-troposphere coupling regarding the varied downward impact of SSWs, using the framework of weather regimes around the onset time of the SSW. This is a novel approach, and the results would have potential implications for subseasonal forecasters and decision makers. It has great potential.

I have spent some time considering the revisions already undertaken, and the comments of the other two reviewers on the first version of the manuscript alongside the authors' responses. I commend the authors for their efforts, particularly in terms of constructing statistical arguments for their results with such a small sample size. However, I am unconvinced that the paper fully discusses its results in the context of other diagnostics of stratosphere-troposphere coupling, and it occasionally lacks clarity in its methods or reasoning. I would argue that using the day 0 regime remains a key issue, as I outline below. I believe that once the authors address these points, the paper will be very useful and serve as a good starting point for further studies using this framework. My recommendation is major revision.

*We would like to thank the reviewer for their thoughtful and thorough evaluation of our manuscript and their helpful comments. We are addressing all points below in the detailed revision.*

**Major comments:**

Classification by day 0 regime: A significant sticking point of this study, raised by previous reviews, is the use of the day 0 classification as a basis. Figure 1 and Table 1 shows there are cases when the day 0 regime is different to the dominant regime in the 10 days leading up to the SSW (e.g. 1980-02-29 classified as "no regime" but right up to onset was in EuBL). I would argue that using day 0 is the main door by which this paper opens itself to criticism. There is no physical reason to just pick day 0, especially seeing as there is no persistence criterion on the regime assignment which, by inspection of Figure 1, shows occasional jumps between regimes (i.e. day 0 could be in a different regime to both day -1 and day +1). Indeed, it is only required that the regime is "active" for 6-hours, which seems a concerningly small time period. These aspects could cause confusion in a real-world forecast scenario, something I can see this paper being used for. In most cases, the day 0 regime is largely similar to the modal regime in the 10 days before the U10-60 reversal. The authors must also consider that the reversal of the 10 hPa 60N U-wind is a quasi-arbitrary threshold and the "sudden stratospheric deceleration" or the 65N reversal often occurs slightly earlier (e.g. the 2 January 2019 SSW) so limiting to one day is problematic. A sub-seasonal forecaster will not yet know the exact date of the U reversal, but will have knowledge of the current/dominant regime state. I therefore strongly suggest the authors consider revising the analysis to use the dominant (i.e. modal) regime in the 10 days before, or the 5 days either side, of the U reversal, or address this issue in some other way.

*Thank you for this constructive comment. We realize that it might not have been sufficiently clear in the manuscript that the regime classification used here is indeed based on a persistence criterion. We here explain again in brevity the regime definition and have further improved the relevant text in the manuscript (lines 102-114 in section 2.1, lines 159-160 in section 3): The regime assignment is based on the projection (in physical space; see Michel and Rivire, 2011) of an instantaneous geopotential height field into a cluster mean derived from a clustering in EOF space (the latter - standard approach - of a mere cluster attribution in EOF space would not necessarily detect persistent states). We normalize this projection into a regime index ($I_{WR}$, in units of standard deviation) to make it comparable for the different regimes. Using $I_{WR}$ we derive weather regime life cycles that fulfil the following key criteria: 1. mean $I_{WR}$ is above 1.0 for at least 5 days. 2. within this period $I_{WR}$ must reach a maximum compared to the $I_{WR}$ for other regimes at least once. 3. $I_{WR}$ must show a mean increase / decrease in a 5 day period prior / after the regime onset / decay. This definition allows simultaneously active weather regime life cycles. That is, a regime 1 can have its onset while a regime 2 is dominant. For attributing a specific date to a regime we therefore only assign the dominant regime, i.e. the regime with the maximum $I_{WR}$ of all active regime life cycles. Thus periods in Figure 1 with alternating dominant regime in short time steps indicate simultaneously active weather regime life cycles. This is admittedly confusing for readers not familiar with our definition and we hope we have clarified this aspect in the revised manuscript. To make our regime attribution at SSW onset (lag of 0 days) even more robust we have followed*

*your advice and we now base the attribution not only on the currently active and dominant regime, but on the maximum mean $I_{WR} \pm 5$ days around the SSW onset of regimes that are active in a time window $\pm 5$ days around SSW onset. The new classification is described on lines 115 - 121. The only change based on this new method of attribution is the SSW in 1980 which is now included into the "EuBL" subset. The overall results and conclusions of our study are not affected by this change. No further attribution changes occurred based on the change to the suggested more robust attribution in a time window around SSW onset, hence we are confident that the regime attribution is robust. We also now exclude the SSW 24.03.2010 following your comment below. We have now included a new table (replacing Table 1 from the former version of the manuscript) to indicate the weather regime index for the WR that has been identified to be dominant around the onset for each SSW.*

L236 and elsewhere: There needs to be more discussion (at the very least) on whether the different tropospheric responses are due to different stratospheric properties in each of the categories. This section of the text introduces the idea, motivated by the previous reviewers, but not adequately. I am concerned that the results overall are just suggesting – in a new way – that split SSWs or longer-lasting SSWs have different tropospheric precursors and are of higher impact than displacements, which is not new. 4 out of the 6 EuBL events are splits, so it is not surprising these have a stronger signal (I am presuming the classification comes from Karpechko et al. 2017 Table 1 [this should be clarified]). The cold anomalies associated with EuBL also recall the results of Lehtonen and Karpechko 2016 JGR. https://doi.org/10.1002/2015JD023860

*We agree with the reviewer, and it is not our intention to state that the tropospheric variability is the only (or, for that matter, the dominant) reason for the type of downward response. While the focus of this study was to investigate the tropospheric response in detail and to clarify its importance to the reader, we agree that stratospheric features will be important for the downward response. The studies investigating stratospheric precursors however suffer from the same problem (as our study) of a small sample size, hence their non-significant results e.g. regarding differences between split and displacement surface impacts. In order to provide more details for the role of precursors, we have updated the Introduction (lines 49 - 65), the Results (lines 179-184, 210 - 214, 244 - 250, 261 - 272), and the Discussion (lines 308 - 314, 339 - 353). The reference for Lehtonen and Karpechko 2016 has been included. The quantification of the respective importance of the stratospheric, local, and upstream precursors will have to be left to a modeling study.*

L137 The periods 60 days before and after may overlap with another SSW. The separation criterion for an SSW is not stated in the methods either (L85) but this timescale implies 60 days. I think this only affects one example, the SSWs in February and March 2010 which are only around 45 days apart. Karpechko et al. 2017 (QRJMS) did not include the March 2010 SSW in their diagnosis of downward impacts. I would suggest the March 2010 SSW is removed.

*The definition for SSW events has now been explained in more detail (section 2.1, see answer to*

*your first comment). The March 2010 event has been removed from the analysis as suggested by the reviewer.*

L155 Blocking in the more general NAE sector (including Scandinavia/Urals) has been suggested in various studies (e.g. Kolstad and Charlton-Perez 2011 https://doi.org/10.1007/s00382-010-0919-7), not just Atlantic ridge (even the cited Martius paper suggests this). So, I am not sure the authors can argue anything particular about the Atlantic ridge.

*We agree that blocking in other regions has been suggested to precede SSW events, though since the weather regimes we use here are limited to the North Atlantic sector, we are not able to compare with these precursors in other locations. The Ural blocking pattern tends to project onto the European blocking pattern, and even more so on the Scandinavian blocking defined for our WR classification. Indeed, EuBL and ScBL have been observed to occur ahead of many SSW events in the beginning of the record as well as towards the end of the record, as has recently been suggested. Martius also suggests blocking in the North Pacific, though we are not able to check this here given that our WRs are defined for the Atlantic only. We have now mentioned additional studies (including the Kolstad and Charlton-Perez study mentioned above) and expanded the discussion of the precursors to SSW events (lines 175 - 184).*

Figure 2: the caption states "note the different y-axis in (a)", but the y-axis in (a) is no different to the others; I would recommend it to be shrunk to better show the variability (presumably this was intended and something went wrong?). The bold lines are also very difficult to discern versus those which are not – could the non-significant parts be dashed, and the bold lines increased in width?

*Thanks for spotting this, the sentence about the y-axis has been removed and the distinction between bold lines and thin lines has been improved.*

Figure 3: if this paper seeks to be as significant and impactful as it aims to be – which I think it can be – then I strongly think this figure should be for the hemispheric-wide anomalies to allow greater intercomparison with other studies of coupling and a greater potential understanding of NAM behaviour. There is little difference with the Fig A3 as the authors state on L195, so I would strongly encourage swapping 3 and A3 with a continued emphasis that the NAE sector dominates (not surprising given the NAO-AO relationship). Additionally, I think the caption for this figure has the hatching and stippling backwards – should it not be "Stippling (hatching) indicates ... overlap by less than 25% (10%)."?

*Thank you for spotting this, stippling and hatching were indeed backwards in the caption. This has been corrected.*

*We understand your reasoning about the hemispheric versus North Atlantic view, but decided to keep Figure 3 in the main text, as the focus of the paper is on the North Atlantic sector. We often refer to this figure in the main text when we discuss the imprint of the specific regimes on*

*the geopotential height anomalies and would therefore prefer to keep it there. Since the Appendix is part of the manuscript in WCD (rather than a separate document) we are not worried that potential interested readers may miss figure A3, which is described in the main text. We have now further clarified that the figure refers to the North Atlantic sector in the figure caption, and we now also refer to figure A3 (the full longitude version of this figure) in the caption of figure 3.*

Figure 4: is any significance testing performed? The caption does not state it, nor is any stippling shown. What base period are the anomalies computed from, 1979-2019? Daily? Moreover, have the authors considered detrending 2m temperature to remove the strong warming signal, especially at higher latitudes?

*We have added a paragraph in the Methods section specifying the definition of the anomalies (lines 153 - 156). The anomalies are computed from a smoothed calendar day climatology over the 1979 - 2019 period using a 21-day running mean filter. As per your recommendation we have detrended the temperature anomalies to take the warming into account. This does, however, not yield significantly different results. In addition, we have added stippling and hatching indicating significance levels of temperature anomalies at the 25 % and 10 % levels, respectively, using the same testing procedure as for the plots in Fig. 3. This has been indicated in the figure caption.*

**Minor comments:**

L26 (and intro generally): there needs to be greater clarity between the NAO defined as an EOF and the regime-based NAO for readers less familiar with regimes.

*We have added additional clarification on the definition of the NAO in the introduction (lines 28 - 35).*

L34: A further relevant reference would be Matthias and Kretschmer, 2020 MWR
https://doi.org/10.1175/MWR-D-19-0339.1

*This reference has been added to the manuscript as suggested (line 42).*

L38 The use of "deterministic" always strikes me as being a bit ambiguous; whether "deterministic" lead-times are the same in the stratosphere (because the timescales are longer). I would suggest being more quantitative. 15 days?

*We have modified this sentence and added additional references in order to clarify this (lines 46 - 48).*

L40 The opening sentence of this paragraph essentially repeats what is said earlier in the introduction.

*This sentence has been re-written to avoid duplication (line 49).*

L50 "..indicators of THE downward impact OF SSW events."
*This has been corrected as suggested by the reviewer (lines 62 - 63).*

L61 No need for "often", i.e. "the stratosphere is ONLY one possible forcing"
*This has been corrected as suggested by the reviewer (lines 73-74).*

L68 "in the NAE region CAN HELP US to understand"
*This has been corrected as suggested by the reviewer (line 77).*

L82 The phrase "is used for all figures". I would suggest replacing "figures" with "analysis".
*This sentence has been re-formulated (line 91).*

L83 There needs to be some additional specifics on the dataset. Is it used at native 0.75 degree resolution, 6-hourly? Is the reversal required to be daily-mean? This also links with Figure 1 – the y-axis labels have _00 on them; if this is not meaningful, it should be removed.
*The hourly part of the y-axis labels (_00) has been removed in Figure 1 and the definition of the SSW central dates has been clarified (section 2.1, lines 91-101). The grid resolution has been clarified for SSW events (line 92) and for the weather regimes (line 103).*

L83 November to March is not midwinter, it is extended winter. Additionally, by including March, final warmings are potentially included (e.g. March 2016) yet this is not listed. The authors should specify how final warmings have been excluded (e.g. Butler and Gerber 2018 https://journals.ametsoc.org/doi/10.1175/JCLI-D-17-0648.1)
*We have removed 'midwinter'. We have also extended the description of the definition of SSW events and we now explain how final warmings are treated in section 2.1 (lines 95 - 98). Since the Butler and Gerber 2018 description of the definition for final warmings contains an error we have now included the explicit definition for final warmings again here, which yields the same events as listed in the Table from Butler and Gerber 2018.*

L86 The definition of a regime as a "quasi-stationary large scale flow pattern" has already been included, so this is superfluous.
*This has been removed as suggested.*

L87 Why are the seven year-round regimes used, since this study is based in wintertime? This should be addressed. What if the classic 4 Cassou regimes were used? Has this been considered?
*The fact that the regimes can be used year-round is seen as a benefit, i.e. seasonal independence. The main reason for using the 7 regimes is that the 4 regimes cannot distinguish all of the variability after SSW events as shown in earlier work (e.g. Beerli and Grams, 2019). In particular, as described in detail in the Discussion, EuBL and GL often occur around the onset of the SSW*

*event, but they have different associated surface temperature signatures. These weather regimes are not possible to distinguish in their clarity in an approach limited to 4 regimes and without the "no regime" classification, which would dilute the clear signals. This has now been clarified in the discussion (lines 317 - 327).*

L88 insert "tropospheric flow conditions IN THE NORTH ATLANTIC at their onset"
*This has been added as suggested (line 116).*

L105 Minor quibble, but I found the readability of the abbreviations to be difficult. Greenland Blocking does not have blocking in its acronym. The authors may wish to consider renaming it to GBL, though I leave this up to them. GL looks like Greenland Low, given than the other blocking patterns have BL.
*This is a good point raised by the reviewer, but in order to keep the acronyms consistent with the acronyms that have been established in the previous literature we have kept the acronym.*

L110 This is phrased too much like a result. I understand the purpose of the sentences here and it links well with my earlier comment on EOF NAO vs. regimes needing more clarity. The authors should consider rephrasing "they are supressed after a weakening of the polar vortex", seeing as this behaviour is what the paper is trying to identify.
*These two sentences have been modified to limit them to the relationship of the NAO with the seven regimes, leaving the link to the stratosphere for the Results section (lines 128 - 129).*

L114 I would argue SSWs are "RELATIVELY rare" as one every 2 winters is much more common than some phenomena. There is also no need to repeat the number of SSWs in ERA-Interim here as it is stated in the methods.
*This has been rephrased (lines 131).*

L115 What does "robust" mean in this context in addition to significant? It is ambiguous.
*We agree that it is not yet clear what we mean by robust in this place as this notion is introduced later. Hence, we have removed the word "robust" as it was not needed here. In general, we consider anomalies to be robust if their amplitude exceeds the width of the confidence interval obtained from a random resampling.*

L182 Why is the response deemed "missing"?
*We agree with the reviewer that the wording might be confusing here. We have updated it to compare to the classification of downward SSW events in Karpechko et al, 2017 (lines 211 - 214).*

L243 To avoid repeating the word anomalies, "composites of anomalies in 2m temperature and 500 hPa geopotential height".

*This has been updated as suggested by the reviewer (lines 276 - 277).*

L295 The phrasing "stratospheric nudging" is not really appropriate given this is not a results of a nudging experiment.
*We have replaced 'nudging' by 'forcing' (lines 334).*

L299 Word order "...the weather regimes considered here"
*This has been updated as suggested by the reviewer (line 338).*

Figure A2: the y-axis is not different to the others as the caption states.
*Thanks for spotting this, this sentence has been removed.*

**The role of North Atlantic-European weather regimes in the surface impact of sudden stratospheric warming events**

Daniela I.V. Domeisen[1,*], Christian M. Grams[2,*], and Lukas Papritz[1,*]

[1]Institute for Atmospheric and Climate Science, ETH Zürich, Switzerland
[2]Institute of Meteorology and Climate Research (IMK-TRO), Department Troposphere Research, Karlsruhe Institute of Technology (KIT), Karlsruhe, Germany
[*]Equally contributing authors

**Correspondence:** Daniela I.V. Domeisen (daniela.domeisen@env.ethz.ch)

**Abstract.** Sudden stratospheric warming (SSW) events can significantly impact tropospheric weather for a period of several weeks, in particular in the North Atlantic European (NAE) region. While the stratospheric forcing often projects onto the North Atlantic Oscillation (NAO), the tropospheric response to SSW events, if any, is highly variable and it remains an open question what determines the existence, location, timing, and strength of the downward impact. We here explore how the variable tropospheric response to SSW events in the NAE region can be characterised in terms of a refined set of seven weather regimes and if the tropospheric flow in the North Atlantic region **around** the onset of SSW events is an indicator of the subsequent downward impact. The weather regime analysis reveals the Greenland blocking (GL) and Atlantic Trough (AT) regimes as the most frequent large-scale patterns in the weeks following an SSW. While the GL regime is dominated by high pressure over Greenland, AT is dominated by a southeastward shifted storm track in the North Atlantic. The flow evolution associated with GL and the associated cold conditions over Europe in the weeks following an SSW occur most frequently if a blocking situation over western Europe and the North Sea (European Blocking) prevailed **around the** SSW onset. In contrast, an AT regime associated with mild conditions **over Europe** is more likely **following the SSW event** if GL occurs already at SSW onset. For the remaining tropospheric flow regimes during SSW onset we cannot identify a dominant flow evolution. Although it remains unclear what causes these relationships, the results suggest that specific tropospheric states **in the days around** the onset of the SSW are an indicator of the subsequent tropospheric flow evolution in the aftermath of an SSW, which could provide crucial guidance for subseasonal prediction.

*Copyright statement.* TEXT

**1 Introduction**

Sudden stratospheric warming **(SSW)** events can have a significant impact on the tropospheric large-scale circulation and hence on surface weather (Baldwin and Dunkerton, 2001). **While a causal downward link from the stratosphere after SSW events has been confirmed in idealized experiments (e.g. Gerber et al., 2009), a robust quantification** of the downward impact

of SSWs in observational data remains challenging. First of all, the number of SSWs in the record of satellite-era reanalysis is small (26 events from 1979 - 2019), while the case-to-case variability in terms of their tropospheric impact is large. Second, the internal variability of the troposphere itself is high, such that it can mask a stratospheric influence. Predicting if, when, and where a downward impact from SSW events will occur is therefore not straightforward, yet a better prediction of the type and timing of a downward impact would significantly benefit a wide range of users.

The tropospheric impact of SSW events is communicated by a range of mechanisms including synoptic and planetary-scale waves (e.g. Song and Robinson, 2004; Domeisen et al., 2013; Hitchcock and Simpson, 2014; Smith and Scott, 2016). **The subsequent tropospheric variability in the North Atlantic-European (NAE) region is often characterized in terms of the bimodal North Atlantic Oscillation (NAO), commonly defined through a station-based index (Cropper et al., 2015; Domeisen et al., 2018), or by the first empirical orthogonal function (EOF) of geopotential height in the North Atlantic sector. Furthermore, multi-modal weather regime classifications based on k-means clustering of the leading EOFs in the North Atlantic sector tend to denote two out of several weather regimes as the positive and negative phases of the NAO given the similarity of their spatial patterns to the bimodal NAO definition (Michelangeli et al., 1995; Cassou, 2008; Ferranti et al., 2015; Charlton-Perez et al., 2018). After SSW events the NAE region tends to exhibit more persistent states of the negative phase of the NAO** (NAO-, Domeisen, 2019), as well as more frequent transitions towards NAO- and fewer away from NAO- (Charlton-Perez et al., 2018). NAO- is associated with enhanced meridional air mass exchanges, in particular, more cold air outbreaks in Northern Europe but fewer over the Nordic Seas (Kolstad et al., 2010; Kretschmer et al., 2018b; Papritz and Grams, 2018; Huang and Tian, 2019), as well as increased precipitation in Southern Europe (Butler et al., 2017; Ayarzagüena et al., 2018). The Pacific sector tends to be less strongly affected in the aftermath of SSW events (Greatbatch et al., 2012; Butler et al., 2017), though the occurrence of wave reflection in the stratosphere can be associated with Pacific blocking (Kodera et al., 2016) and cold spells over **North America (Kretschmer et al., 2018a; Matthias and Kretschmer, 2020).** Given the preferred occurrence and the increased persistence of certain surface signatures **in the NAE region** after SSW events as compared to climatology, medium- to long-range predictability over Europe has been suggested to increase after SSW events (Sigmond et al., 2013; Domeisen et al., 2015; Karpechko, 2015; Butler et al., 2016; Scaife et al., 2016; Jia et al., 2017; Beerli et al., 2017; Butler et al., 2019; Domeisen et al., 2020a). SSW events themselves are often not predictable **beyond a few days to weeks, with high inter-event variability (Taguchi, 2014, 2016; Domeisen et al., 2020b), although probabilistic predictability can be found for longer timescales (Scaife et al., 2016).**

**The analysis of the high case-by-case variability in the tropospheric signature after an SSW event is** further complicated by the fact that there exists a range of different metrics for characterizing the downward impact, with each definition yielding a different set and number of SSW events with apparent surface impacts. In particular, the occurrence and type of downward impact has been investigated with respect to the SSW **geometry, i.e., split versus displacement events (Charlton and Polvani, 2007; Mitchell et al., 2013; Maycock and Hitchcock, 2015; Seviour et al., 2016; Lehtonen and Karpechko, 2016),** though no statistically robust differences with respect to wave geometry emerge in the tropospheric response. **In addition, it has been suggested that precursors to SSW events with a downward influence differ from SSWs without such a tropospheric impact in terms of strength and location (Nakagawa and Yamazaki, 2006; Domeisen, 2019; Zhang et al.,**

**2019), in particular with respect to forcing over Eurasia (White et al., 2019; Tyrrell et al., 2019; Peings, 2019). Furthermore, the** evolution of the stratosphere - troposphere system following the SSW (Kodera et al., 2016) and in particular the persistence of the lower stratospheric response after the SSW event (Hitchcock et al., 2013a; Karpechko et al., 2017; Runde et al., 2016; Polichtchouk et al., 2018) **have been found to determine the existence and type of a downward response.** These studies use indices for the downward effect that are based on exclusively stratospheric or a combination of stratospheric and tropospheric indicators. **For comparison,** in this study we will investigate purely tropospheric indicators **of the downward impact of** SSW events. Definitions of a downward impact using tropospheric indicators are generally based on large-scale circulation indices such as the NAO (Charlton-Perez et al., 2018; Domeisen, 2019) or tropospheric jet location (Garfinkel et al., 2013; Afargan-Gerstman and Domeisen, 2020; Maycock et al., 2020).

**Furthermore,** remote forcing can affect both the stratosphere and the troposphere, and thereby either mask or strengthen the downward response from the stratosphere. A range of tropical remote connections can impact the NAE region through both a tropospheric and a stratospheric pathway (Attard et al., 2019), such as the Quasi-Biennial Oscillation (QBO) (Gray et al., 2018; Andrews et al., 2019), the Madden-Julian Oscillation (MJO) (Garfinkel et al., 2014; Barnes et al., 2019), and El Niño Southern Oscillation (ENSO) (Jiménez-Esteve and Domeisen, 2018; Domeisen et al., 2019), in addition to extratropical tropospheric forcing in the North Pacific (Honda and Nakamura, 2001; Sun and Tan, 2013; Drouard et al., 2013), Arctic sea ice (Sun et al., 2015), and snow cover in Eurasia (Cohen et al., 2014). It therefore has to be kept in mind that the stratosphere **is only one possible forcing** of the troposphere.

Given the large variability of the tropospheric flow evolution following SSW events and the influence of additional remote factors mentioned above, the prediction of the SSW response in the troposphere remains difficult for an individual event, despite the general shift towards NAO negative conditions in a statistical sense. The goal of this study is to investigate if tropospheric flow regimes in the NAE region **can help us understand** the variability of the SSW response in the observational record. More specifically, we here address the question if the tropospheric flow evolution in the NAE region after an SSW is statistically different from that without an SSW using seven weather regimes in the NAE region. Weather regimes are quasi-stationary, recurrent, and persistent patterns of the large-scale extratropical circulation (e.g. Michelangeli et al., 1995). While many studies showed that there are preferred transitions between different regimes, internal tropospheric variability is high and a regime onset often occurs on short timescales (e.g. Vautard, 1990; Michel and Rivière, 2011). Therefore predictability due to regimes arises from regime persistence on time scales of several days rather than typical regime sequences over several weeks. However, recent work revealed important shifts of regime occurrence and transition probabilities between regimes on subseasonal time scales of several weeks dependent on the external forcing such as the stratospheric polar vortex state (Charlton-Perez et al., 2018; Papritz and Grams, 2018; Beerli and Grams, 2019). This motivates the study at hand aiming at investigating if the variability in the tropospheric flow evolution following SSW events can be characterised in terms of the weather regime **around the** SSW onset.

**2 Data and Methods**

 ### 2.1 Data and Classifications

ERA-interim reanalysis (Dee et al., 2011) from 1979 to **2019 is the data basis for this study**. **The SSW events are defined based on daily mean data at the native ERA-interim horizontal grid resolution. The SSW central dates are defined as the first day of easterly zonal mean zonal winds at 10 hPa and 60°N between the 1st of December and the 31st of March. Events have to be preceded by at least 20 consecutive days of daily mean westerly winds. If an event fulfills the criterion for both a SSW and a final warming event, it is excluded from the analysis. Final warming events are defined as the first day of the year when the zonal mean zonal winds at 10 hPa and 60°N reverse and do not return to westerly for more than 10 consecutive days. These SSW central dates agree with Table 2 in Butler et al. (2017), which provides central dates up to 2013, and are defined using the same criterion thereafter. The central dates for the more recent SSW events are 12-Feb-2018 and 02-Jan-2019 (see Table 1). Following Karpechko et al. (2017) the event on 24-03-2010 has been excluded to avoid an overlap with the aftermath of the SSW event on 09-02-2010. This yields 25** SSW events for the period 1979-2019.

The tropospheric flow over the NAE region is described **in terms of seven year-round weather regimes** defined in Grams et al. (2017) based on six-hourly data for the period 1979 - **2019 using 1.0° horizontal resolution (Figure A1).** As for the canonical seasonal definition using four regimes (e.g. Michelangeli et al., 1995; Michel and Rivière, 2011; Ferranti et al., 2015; Charlton-Perez et al., 2018), the mean patterns of the seven regimes are based on a k-means clustering in the phase space spanned by the leading seven EOFs (explaining 76% of the variance) of 10-day low-pass filtered 500 hPa geopotential height anomalies. In addition, **we employ a normalized projection (weather regime index $I_{WR}$) following Michel and Rivière (2011) for each of the seven regimes to define objective and persistent** weather regime life cycles and for a filtering of time steps without a clear regime structure ("no regime" category). In essence, an active life cycle requires an $I_{WR}$ above a certain threshold for at least 5 consecutive days (**minimum persistence of** an active regime life cycle) and a continuous increase/decrease during the onset/decay phases (see methods of Grams et al., 2017, for details). As different life cycles can be active simultaneously, in particular during the onset and decay phases, individual days are attributed to a specific regime life cycle only if $I_{WR}$ is also the maximum of all $I_{WR}$. The life cycle definition allows for a continuous extension of the weather regime attribution to more recent data without repeating the EOF analysis and clustering (here done for the years 2016-2019).

**We use this weather regime classification to stratify SSW events according to the large-scale tropospheric flow conditions in the North Atlantic around their onset (see Table 1). To do so, we select for each SSW the "dominant" weather regime that is active during at least one 6-hourly time step in a time window $\pm 5$ days around the onset day (at 0 UTC) of the SSW. We consider a weather regime to be dominant if the mean $I_{WR}$ in the time window $\pm 5$ days around the onset reaches a maximum compared to other active regime life cycles. Manual inspection of the 25 considered SSW events confirms the unambiguousness of this approach. An identified weather regime is required to be dominant for a minimum of 3 days in the considered 10-day period around the SSW central date.**

Three of the seven regimes are dominated by a cyclonic 500 hPa geopotential height anomaly ("cyclonic regimes"; cf. Figs. A1a-c): the Atlantic Trough (AT) regime with cyclonic activity shifted towards western Europe, the Zonal regime (ZO), and the Scandinavian Trough (ScTr) regime. The remaining four regimes are dominated by a positive geopotential height anomaly and are referred to as "blocked regimes" (Figs. A1d-g): Atlantic Ridge (AR), European Blocking (EuBL), Scandinavian Blocking (ScBL), and Greenland Blocking (GL).

A potential modulation of the frequency of occurrence of the seven regimes can be understood in terms of the link between the respective regimes and the NAO (Beerli and Grams, 2019, their Figs. 2, 6). **While ZO and ScTr project onto NAO+, GL strongly projects onto NAO-. EuBL and AT do not project strongly onto either NAO phase.**

**2.2 Statistical testing**

**Since SSW events only occur roughly every second winter (Butler et al., 2017),** the subsequent stratification according to tropospheric flow conditions requires careful statistical testing to extract significant results that are distinct from sampling uncertainty. The overarching questions we address in this study are whether after SSWs the tropospheric flow evolution is different from situations without an SSW and to what extent this depends on the tropospheric state at the time of the SSW. To investigate **these questions,** we consider subsamples of all SSWs. In all cases the relevant null hypothesis is that the flow evolution after SSWs is indistinguishable from that occurring in the absence of an SSW. The testing procedure, thus, comprises the following two steps:

1. First, we assess the *robustness* of the samples by performing a Monte Carlo resampling. For that purpose, we resample the original samples 100 times with repetitions. The number of random samples is chosen according to the maximum number of possible combinations with repetitions of the smallest subset of SSW events that will be considered in this study ($N = 5$ events corresponding to 126 independent combinations). This yields confidence intervals, estimating the uncertainty inherent in each sample. Due to the small sample size, these confidence intervals are relatively large.

2. Second, we compute 1000 random samples of the same size as the original sample but for random periods with the same weather regime at the central date but no SSW occurring within $\pm 60$ days, yielding estimates of the distributions in the absence of SSWs. Prescribing the same weather regime at the central date for the random samples filters out signals which might result from regime persistence or preferred regime transitions independent of external forcings. Testing for *significance* is done by comparing the confidence intervals and distributions obtained from the random samples for overlap.

Applying this method to **anomalies of geopotential height and 2m temperature**, we consider anomalies as robust if the width of the confidence interval is smaller than the amplitude of the anomaly. In addition, the sample mean is significant at, e.g., the 10% level, if the confidence intervals overlap by less than 10% with the Monte Carlo distribution. A similar procedure is applied to test significance of lagged weather regime occurrence.

**Anomalies of geopotential height are defined with respect to the climatological (1979 - 2019) 21-day running mean. In order to remove the background warming, which is particularly pronounced at high latitudes, we consider detrended**

**Table 1.** Weather regime attribution around the onset of SSW events: SSW date, attributed regime and mean weather regime index ($\overline{I}_w$, with $w \in AT, ZO, ScTr, AR, EuBL, ScBL, GL$) for the attributed regime for the period $\pm$ 5 days around SSW onset. (*) indicates that the event has been excluded from the subsequent analysis, for details see section 2.1.

| SSW central date | attributed regime | $\overline{I}_w$ |
|---|---|---|
| 22.02.1979 | EuBL | 1.41 |
| 29.02.1980 | EuBL | 0.91 |
| 04.03.1981 | GL | 1.82 |
| 04.12.1981 | AR | 2.18 |
| 24.02.1984 | EuBL | 1.01 |
| 01.01.1985 | EuBL | 0.74 |
| 23.01.1987 | AR | 1.52 |
| 08.12.1987 | GL | 1.56 |
| 14.03.1988 | AT (cyclonic) | 0.48 |
| 21.02.1989 | ZO (cyclonic) | 1.74 |
| 15.12.1998 | ZO (cyclonic) | 1.01 |
| 26.02.1999 | ScTr (cyclonic) | 1.37 |
| 20.03.2000 | ScTr (cyclonic) | 0.91 |
| 11.02.2001 | EuBL | 0.47 |
| 30.12.2001 | GL | 1.31 |
| 18.01.2003 | no | – |
| 05.01.2004 | no | – |
| 21.01.2006 | EuBL | 1.14 |
| 24.02.2007 | GL | 1.36 |
| 22.02.2008 | ZO (cyclonic) | 1.32 |
| 24.01.2009 | AT (cyclonic) | 1.86 |
| 09.02.2010 | GL | 2.46 |
| 24.03.2010* | AT (cyclonic) | 1.20 |
| 07.01.2013 | EuBL | 1.09 |
| 12.02.2018 | ZO (cyclonic) | 1.48 |
| 02.01.2019 | AR | 1.52 |

155   anomalies of 2m temperature. For that purpose, we use as the climatology a centered 9-year mean instead of the entire study period. Note that at the beginning and end of the study period the first and last 9 years are used, respectively.

**3 Weather regimes during SSW events**

As a first step, we evaluate the sequence of weather regimes from 60 days before to 60 days after an SSW for all 26 SSW cases during 1979-2019 (Fig. 1, cf. Table 1). **Note that Fig. 1 shows the dominant persistent regime, so that alternating regimes in a time window shorter than the persistence criterion of 5 days indicate simultaneously active regime life cycles (see Section 2.1 for details).** This figure suggests a preferred occurrence of AT (purple) and GL (blue) during the weeks after an SSW compared to the weeks before. This is further emphasized by the 5-day running mean of the anomalous frequency of

[Figure]

**Figure 1.** The sequence of the dominant weather regimes (colors indicated in legend) for -60 to +60 days with respect to the onset **for all** 26 SSW events (lag 0) between 1979 and 2019. The central dates of the SSW events are indicated on the left. **(*) indicates that the event has been excluded from the subsequent analysis, for details see section 2.1.**

weather regimes around SSW events, which provides a more complete overview over the modulation of regime frequencies after SSWs (Fig. 2). We show the 5-day running mean frequency anomaly to account for the 5-day minimum duration of an active regime life cycle. Different from the testing procedure outlined in Section 2.2, we here consider the distribution of lagged 5-day mean frequencies by selecting for each day in the original sample a random day $\pm15$ days around the original day of year but from a different winter. In addition, the random day must exhibit the same weather regime as the original day to replicate

potential regime-dependence. We then compute the mean lagged weather regime frequency for each random sample as for the original sample and test for significance at the 10% level (bold). For reference, we show the absolute frequencies of weather regimes in Fig. A2.

GL and EuBL are the most prominent regimes **around** the onset of SSW events with a 5-day mean frequency of **around 19% and 21%**, respectively (Figure A2a). The frequency of EuBL is significantly enhanced from 5 days prior until the onset of the SSW (Figure 2a), in agreement with Woollings et al. (2010) and Nishii et al. (2011). The cyclonic regimes ZO, ScTr, as well as the blocked regimes AR and ScBL tend to be suppressed at the time of SSW events. This is consistent with the strong projection of the ZO and ScTr regimes onto NAO+, which also tends to be suppressed after SSW events (Charlton-Perez et al., 2018). On the other hand, AR (yellow, significant peak around lag -20 to -10) and the related ScTr (orange, significant **around lag -10**) regimes are more frequent in the period 1-3 weeks before the SSW onset. The prominence of AR around 15 days before the onset of an SSW event agrees with the suggested precursor role of blocking over the Atlantic before SSW events (Martius et al., 2009). **Furthermore, blocking over the Ural region in Eurasia has been suggested as a precursor to SSW events Kolstad and Charlton-Perez (2011); Peings (2019); White et al. (2019). The Ural blocking precursor projects onto the EuBL and especially the ScBL regimes, which are also found to show significant positive anomalies of occurrence within the 3 weeks before SSW events (Fig. 2a). It is well known that precursors in the North Pacific also tend to be prevalent before SSW events, e.g. Garfinkel et al. (2012); Lehtonen and Karpechko (2016), though these are not possible to identify with the present analysis, which is focused on the NAE region.** After the SSW onset, AT frequencies are significantly enhanced, peaking at **around 20%** after 7 days (Figure A2a) corresponding to a frequency anomaly of **around 12%** for the same lag (Figure 2a). Thereafter, GL (lag 12 to **40 days**) and AT (lag 17-35 days) are the most likely weather regimes with enhanced frequency anomalies of up to 15% (Fig. 2a), while in absolute terms frequencies for both are around 20-25% and none of the two clearly dominates (Figure A2a). This **dominant occurrence of both GL and AT after SSW events** obscures the potential tropospheric impact of an SSW in a composite, as AT and GL trigger contrasting large-scale weather conditions (mild and windy for AT, cold and calm for GL) for large parts of Europe (Beerli and Grams, 2019).

We now sub-divide the **25 SSW events** with respect to the weather regime that dominates during the **10 days around the** SSW onset: GL (5 cases), EuBL **(7 cases), and the cyclonic regimes (ZO, ScTr, AT; 8 cases). The remaining 5 cases either have no clear regime signature (no-regime, 2** events) or are associated with AR (3 events) at their onset. Because of the small sample size, we do not consider these cases here. For the GL subset (Fig. 2b / A2b), all other regimes are subsequently suppressed except for AT and EuBL. The frequency of GL itself drops immediately after the **SSW, reaching values below 10% around a lag of 20 days (Fig. A2b)** – far below its climatological mean frequency. AT, and to a lesser degree also EuBL, become significantly more frequent immediately after the SSW until about a lag of 10 days, reaching absolute frequencies of 35% and 20%, respectively (Fig. A2b). After a period with no clear regime assignment, AT becomes the dominant regime starting at lag 18 days with anomalous frequencies above 40% (Fig. 2b), peaking above 50% absolute frequency about 23 days after the SSW and remaining significantly enhanced until a lag of 33 days (Fig. A2b). From lag 25 days until lag 40 days, EuBL becomes significantly enhanced peaking at 40% absolute frequency around lag 30 days.

[Figure]

**Figure 2.** 5-day running mean of the *anomalous* frequency of weather regimes centred on the onset of the SSW event (lag 0) relative to the mean of the climatological distribution for (a) all SSW events and (b-d) conditional on the dominant weather regime **around** lag 0: (b) Greenland blocking, (c) European blocking, and (d) cyclonic regimes (ZO, AT, and ScTr). The 5-day mean frequencies are computed from 6-hourly weather regime data for lags of -60 days to 60 days. Note that anomalous frequencies at lag 0 in c, d are - by construction - close to zero as the same regime is prescribed for computing the mean from the 1000 Monte-Carlo samples. The bold parts of the lines indicate significant deviations from climatology (see text for details).

For the EuBL subset (Fig. 2c / A2c), the subsequent regime frequencies are quite different to GL **around** the onset of an SSW. First, the **frequency of AR is** significantly enhanced directly after the SSW, with peaks at **30%** absolute frequency at lag 10 days. This is then followed by a period of preferred occurrence of GL (**lag 15 to 25 days) and AT (lag 21 to 32 days) with**

205 **an absolute frequency reaching up to 45% and 35%, respectively**. The dominance of GL from lag **33** to 45 days (**above 45**% peak frequency) is particularly striking.

Cyclonic regimes **around** the time of the SSW (Fig. 2d / A2d) exhibit a less prominent regime frequency modulation after an SSW compared to the EuBL and GL subsets. Still, GL (lag **5**-35 days), AR (lag **24-31** days), and AT (lag 25-**37** days) are significantly enhanced, but absolute frequencies **remain around 20-30%.** Note that this corresponds to significantly increased

210 frequencies of 10-20% for these regimes in the considered time windows. However, most often no single regime dominates after an SSW event with a cyclonic regime at lag 0, hinting at cases with **no downward** response after the SSW event. **Of the 8 SSWs with cyclonic weather regimes at lag 0, Karpechko et al. (2017) investigated 7 and classified 5 out of the 7 SSW events as lacking a tropospheric impact. For EuBL and GL around the onset of the SSW, 7 out of 7 and 1 out of 5, respectively, are classified as having a tropospheric impact. The reasons for this will be discussed in the next section.**

Despite the large tropospheric variability in the aftermath of SSW events, the investigation of lagged regime frequencies reveals that (1) the AT and GL regimes are more likely to follow an SSW (as compared to other weather regimes) and (2) that this subsequent modulation is sensitive to the tropospheric flow regime **around** the onset of the SSW. The dominance of EuBL and GL at the time of the SSW onset hints at a significantly more likely GL response (after EuBL at lag 0) vs. AT (after GL at lag 0) after an SSW, respectively. Thus the stratospheric impact on the evolution of the tropospheric flow in the NAE region and hence the associated surface weather may be connected to the presence of a particular tropospheric regime **around** the onset of the SSW.

**4  Temporal Evolution of the Downward Impact**

We focus in the following on the modulation of stratosphere-troposphere coupling for the previously discussed sets of SSWs. For that purpose, we evaluate the temporal evolution of standardized geopotential height anomalies averaged over the NAE sector (-80°E to 40°E / 60°N to 90°N) by compositing a given set of SSW events. Using the full hemisphere, that is, the full longitude range instead of the here used **longitude sector** over the North Atlantic, yields the same qualitative results due to the strong imprint of the anomalies induced by the SSW in the NAE sector (Fig. A3).

Compositing all SSW events (Fig. 3a) yields the classical dripping paint plot of Baldwin and Dunkerton (2001, their Fig. 2). Qualitative differences to the figure from Baldwin and Dunkerton (2001) are due to the different variable (geopotential height in our study vs NAM) and the number of events (**25** in our study vs 18) for a different time period (1979-2019 in our study vs 1958-1999). When compositing all SSW events, the downward impact between 10 to 60 days after the SSW onset is robust at the 25 % but not the 10 % level (see Fig. A4a). Together with the relatively weak amplitude of the anomalies, this reflects the large case-to-case variability in the tropospheric impact of SSWs. **Despite the low robustnes, the anomaly around a lag of 15 days is unlikely to be obtained from a random sampling as evident from the less than 10 % overlap between the confidence and random distributions (Fig. 3a).** This suggests that in the aftermath of an SSW (lag 15 - 25 days), indeed positive geopotential height anomalies over the NAE sector are significantly more likely than in the absence of an SSW.

SSW events that occur during GL (Fig. 3b) are associated with an immediate, strongly positive anomaly in the troposphere. Consistent with Fig. A2b, when GL is present **around** the onset of the SSW, GL or AR are often already present before the SSW event, which is likely the cause of the positive tropospheric geopotential height anomalies several days prior to the event. Notably, there are no significant and robust (cf. Figs. 3b and A4b) anomalies after 10 days of the onset of the SSW except for a weak negative geopotential height anomaly after 20 days **(significant at the 25 % level),** indicating a cyclonic flow regime in the NAE region. This is consistent with the significantly enhanced likelihood for the occurrence of the AT regime at this lag (Fig. A2b). Note that both the immediate positive geopotential height anomalies and the weak tropospheric anomalies in the aftermath of the event are not the result of cancellations in the composites but are rather typical across cases. **In fact, 4 out of the here identified 5 GL events have been classified by Karpechko et al. (2017) as having no downward impact.**

**For EuBL around the onset of the SSW event, a robust (10 % level) positive tropospheric anomaly can be observed at the time of the SSW (Figure A4c). This anomaly is not significant (Figure 3c), reflecting that it is not different from**

[Figure]

**Figure 3.** Standardized geopotential height anomalies for the **North Atlantic** sector [-80°E to 40°E / 60°N to 90°N] for (a) all SSW events, and (b - d) sub-divided by the weather regime that is dominant **around** the onset of the SSW event as indicated in the panel titles. **Stippling (hatching)** indicates that the confidence intervals and the random distributions overlap by less than 25% (10%). **Figure A3 shows a version of this figure for the full longitude range.**

**generic anomalies during EuBL. However,** robust, significant, and strongly positive geopotential height anomalies are present in the troposphere at lags of 15 - 20 and 30 - 55 days after the SSW event. **This is consistent with the classification of all**
250  **of the here defined 7 EuBL events as having a tropospheric impact in Karpechko et al. (2017).** These positive anomalies are consistent with the finding that first AR and then GL are much more likely in the aftermath of an SSW with EuBL **around** lag 0 (compare to Fig. A2c). Furthermore, comparing to the panel for all SSW events (Fig. 3a) indicates that the EuBL cases dominate the perceived downward response in the canonical response for SSW events.

During cyclonic regimes **around** the onset of the SSW, there is no substantial tropospheric anomaly in the NAE region at the
255  time of the SSW, but a positive albeit weak anomaly can be observed around days 15 - 20 after the SSW event (Fig. 3d). This anomaly is not robust at the **25 % level, but it is significantly different from a random sample at the 25 % level (Figure A4d)**. Several SSWs with a cyclonic regime **around** the onset are followed by GL at a longer lag (Fig. A2d), thus likely causing these anomalies. Still, the GL **absolute frequencies remain below 30% (Figure A2)**. These findings and the small amplitude of the anomalies suggest that the variability in the tropospheric flow evolution after SSWs is large after a cyclonic regime at
260  lag 0, which is also confirmed by the inspection of individual cases (not shown).

The question arises whether other factors might contribute to the differing **tropospheric** evolution in the aftermath of the SSW event. In particular, a differing amplitude and persistence of the lower stratospheric anomaly can be observed in Fig. 3 between the different composites. Events with EuBL **around** the onset **and a strong downward impact tend to have a**

**longer stratospheric persistence, but an equally long persistence can be observed for cyclonic regimes around the onset of the SSW, with little downward impact.** The five SSW events associated with GL have a shorter-lived lower stratospheric response. As events with a persistent lower stratospheric response are often associated with so-called polar jet oscillation (PJO) events (Kuroda and Kodera, 2004; Hitchcock et al., 2013b), a comparison with Table 1 in Karpechko et al. (2017) reveals that 2 out of 5 SSW events with a GL regime (and, respectively, 4 out of **7** EuBL events) **around** the onset **are associated with** a PJO event. While this is not a clear result, it indicates that the shorter (longer) persistence in the lower stratosphere for the SSWs associated with GL (EuBL) may add support to the persistence of the tropospheric response **for several of the events**, but the statistics are too small to provide a clear result. Similarly, 4 out of **7** EuBL events are split events (rather than displacements), while 2 out of 5 GL events are split events, **according to the classification in Karpechko et al. (2017).**

**5   Impact on Surface Weather**

Since each weather regime is associated with characteristic surface weather, the modulation of regime successions in the aftermath of an SSW by the tropospheric state at the time of an SSW might contribute to the marked variability in the surface impact. Hence, we here consider spatial composites **of anomalies of 2m temperature (T2m') and 500 hPa geopotential height** (Z500') for the three groups of SSW events discussed in the previous sections (Fig. 4a-c) and for all SSW events (Fig. 4d) for days 0 to 25 after the SSW (cf. Fig. A5 for days 25 - 50).

During SSWs dominated by GL **around their** onset, initially **strongly positive T2m'** prevail over Greenland and the Canadian Archipelago, whereas western Russia and Scandinavia are anomalously cold, consistent with the anomalous ridge over Greenland and the low geopotential height anomalies over Scandinavia (Fig. 4a). With the subsequent progression of weather regimes - typically towards the cyclonic AT regime or EuBL - mild conditions are established throughout central Europe from a lag of 20 days onwards. This is in stark contrast to the negative NAO phase and the associated cold conditions that are commonly expected as the canonical response to SSWs over Europe (Butler et al., 2017; Kolstad et al., 2010; Domeisen et al., 2020a).

For SSWs that are dominated by EuBL **around their** onset, cold anomalies prevail over Northern Europe, albeit also extending over large parts of central Europe (Fig. 4b). They peak at -4 K to -6 K around lags beyond 20 days, which corresponds well with the occurrence of the GL regime. Note that **negative T2m'** in the composite for all SSWs are much weaker (cf. Fig. 4d). **The associated** retrogression of initial positive Z500' over the eastern North Atlantic to Greenland along with a strengthening of negative Z500' over the southeastern North Atlantic around lag 15-25 days is striking. Furthermore, GL is associated with warm anomalies over Greenland and **eastern Canada**.

Finally, as expected by the varied regime succession for the SSWs with cyclonic regimes at their onset, composite **T2m'** and Z500' are weaker for these events (Fig. 4c). Thus, the canonical response of surface temperature (i.e., the composite for all SSWs, Figure 4d) is the result of averaging over – in important regions opposing – temperature anomalies for SSWs with GL, EuBL, or a cyclonic regime **around** the onset.

[Figure]

**Figure 4.** Surface impact for SSWs with (a) Greenland blocking, (b) European blocking, and (c) cyclonic regimes **around the SSW** onset, as well as (d) for all SSWs. **Shading indicates the composite 2m temperature anomalies with stippling (hatching) indicating significance at the 25 % (10 %) level. Blue contours correspond to geopotential height anomalies at 500 hPa in intervals of 50 gpm. Negative values are dashed. The fields are averaged over 5 days between lags 0 to 25 days with respect to the SSW central date. Note the different scales for temperature in (a-c) and (d). The 2m temperature anomalies are detrended and deseasonalized using 9-year and 21-day running mean filters.**

**6 Summary and Discussion**

This study aimed to shed light on the large case-to-case variability of the tropospheric response to SSW events and their associated surface impacts, as well as the dependence on the tropospheric weather regime **around** the onset of the SSW. To that end, we have exploited in a statistical framework the observational record of the satellite era (1979 - 2019) as represented in the ERA-Interim reanalysis. Our conclusions are as follows:

1. In the aftermath of an SSW event, the tropospheric flow in the NAE region exhibits an evolution that is unlikely to occur in the absence of an SSW. Specifically, positive geopotential height anomalies related to Greenland blocking are statistically more likely to occur after the onset of the SSW than in the absence of an SSW. This is consistent with the expected (canonical) negative NAO response of the troposphere to SSWs (e.g. Charlton-Perez et al., 2018).

2. The significant and robust positive geopotential height anomalies found in the period 10-60 days after SSWs are predominantly the result of SSWs with European blocking **dominating around** their onset. This is manifest for this subset of events in a transition from EuBL to GL that then dominates at lags of 15-20 and 30-55 days after the SSW onset, which is statistically significantly different from the natural progression from EuBL to GL. **These events all correspond to SSWs that have in the literature been classified as having a tropospheric response (e.g. Karpechko et al., 2017).** For other tropospheric regimes at SSW onset the tropospheric response is weaker and less robust and significant.

3. For Greenland blocking at the SSW onset, a weak preference for cyclonic flow regimes around 20-30 days after the SSW is apparent, with an opposite surface response in the aftermath of the SSW as compared to SSW onsets dominated by EuBL. **These events almost exclusively correspond to SSWs that have in the literature been classified as having no tropospheric response.**

4. SSWs that occur during cyclonic weather regimes exhibit a considerably weaker and less significant response **as compared to SSW events associated with EuBL,** with a modestly enhanced likelihood for GL.

Depending on the tropospheric weather regime around the SSW onset different surface signatures result. Specifically, the signature in 2m temperature **resembling the expected canonical NAO- state**, e.g., cold conditions prevailing over much of northern Europe, occurs for the EuBL cases. In contrast, mild temperatures in large parts of Europe are found for SSWs with GL **around** their onset. It is important to distinguish these cases, since although EuBL and GL frequently **(that is, for roughly 50% of all SSWs)** occur **around** the onset of SSW events, they lead to a different subsequent evolution and different associated surface temperatures. In particular, the most common **SSW events exhibit a transition from EuBL (GL) around SSW onset to GL (AT)** around 3-4 weeks after the SSW, respectively, along with their contrasting large-scale weather impacts (Beerli and Grams, 2019). **Note that these differences cannot be identified by using a set limited to 4 weather regimes.** These findings **indicate** that the presence of either a EuBL or GL regime at SSW onset will allow us to disentangle the difference in surface weather, and hence to determine if and when a downward impact of the SSW is expected. This is highly relevant for subseasonal forecasting.

While these findings are limited by the small sample size of **SSW events available in the observational record,** the rigorous statistical testing for significance and robustness performed here suggests that the large case-to-case variability in the tropospheric response to SSWs can be described in terms of NAE weather regimes and may depend on the regime **around the onset of the SSW for many of the observed SSWs.** Our findings confirm that while the stratosphere does not represent the sole forcing of the tropospheric state, for many SSW events it is able to **affect** the tropospheric flow by suppressing some weather regimes and by favoring others, as found in Charlton-Perez et al. (2018). We here in addition show that the

susceptibility of the troposphere to the stratospheric **forcing** depends on the tropospheric state **around** the time of the SSW. Other factors that can modulate the tropospheric response **to SSW events** are the persistence of the temperature anomaly in the lower stratosphere (Hitchcock et al., 2013a; Karpechko et al., 2017; Runde et al., 2016; Polichtchouk et al., 2018), as well as upstream effects in the North Pacific (Afargan-Gerstman and Domeisen, 2020; Jiménez-Esteve and Domeisen, 2018). An analysis of differences in the lower stratospheric persistence for **the weather regimes considered here** did not yield conclusive results, which warrants further studies. **Note that it was not possible in our analysis to fully exclude differences in the stratospheric forcing between SSW events due to the small sample size. In particular, differences in stratospheric behavior, such as vortex geometry or the persistence of the temperature signal in the lower stratosphere, may influence the type and persistence of the downward response. We expect a negligible influence from the vortex geometry, as the differences in the surface signals between split and displacement events tend to be small (Charlton and Polvani, 2007; Mitchell et al., 2013; Maycock and Hitchcock, 2015; Seviour et al., 2016; Lehtonen and Karpechko, 2016), and are also affected by the small sample size. While the persistence of the lower stratospheric response likely affects the persistence and type of the tropospheric signal (Hitchcock et al., 2013a; Karpechko et al., 2017; Runde et al., 2016; Polichtchouk et al., 2018), we did not find a clear correspondence between persistent stratospheric events and tropospheric weather regime evolution. In particular, roughly half of SSW events associated with either EuBL (4 out of 7) or GL (2 out of 5) are associated with a persistent lower stratospheric response (note the small sample size). Hence, we could not find a clear equivalence between tropospheric weather regimes and lower stratospheric persistence.**

**Our goal is to emphasize that the troposphere has a role to play in the downward response of SSW events. The respective contributions of the stratosphere, the state of the troposphere over the North Atlantic, and upstream precursors will subsequently have to be disentangled in a modeling study.** In particular, a model study to quantify the respective contributions to the tropospheric impact of different remote factors in comparison to the role of local North Atlantic variability might shed further light onto the complex role of stratosphere - troposphere coupling for surface weather. However, it is currently not sufficiently known to what extent complex prediction models are able to represent the diversity of tropospheric responses to stratospheric forcing, as this has not been sufficiently tested in models beyond the canonical response and selected case studies. From a preliminary analysis of **subseasonal prediction models** we anticipate large biases and a complex role of the representation of stratosphere - troposphere coupling in prediction models that will be difficult to disentangle. Hence, while state-of-the-art subseasonal prediction systems are often unable to forecast at the time of occurrence of the SSW event if a surface response is to be expected, our findings suggests that the presence or absence – and in fact the timing – of a surface impact following SSW events might in some cases be predictable based on the **dominant weather regime around** the onset of the SSW event. This could significantly improve the subseasonal prediction of tropospheric winter weather **following SSW events** over Europe.

[revised manuscript text omitted]

Domeisen, D. I. V., Badin, G., and Koszalka, I. M.: How Predictable Are the Arctic and North Atlantic Oscillations? Exploring the Variability and Predictability of the Northern Hemisphere, Journal of Climate, 31, 997–1014, 2018.

Domeisen, D. I. V., Garfinkel, C. I., and Butler, A. H.: The Teleconnection of El Niño Southern Oscillation to the Stratosphere, Reviews of Geophysics, 57, https://doi.org/10.1029/2018RG000596, 2019.

Domeisen, D. I. V., Butler, A. H., Charlton-Perez, A. J., Ayarzaguena, B., Baldwin, M. P., Dunn Sigouin, E., Furtado, J. C., Garfinkel, C. I., Hitchcock, P., Karpechko, A. Y., Kim, H., Knight, J., Lang, A. L., Lim, E.-P., Marshall, A., Roff, G., Schwartz, C., Simpson, I. R., Son, S.-W., and Taguchi, M.: The role of the stratosphere in subseasonal to seasonal prediction: 2. Predictability arising from stratosphere - troposphere coupling, Journal of Geophysical Research-Atmospheres, https://doi.org/10.1029/2019JD030923, 2020a.

Domeisen, D. I. V., Butler, A. H., Charlton-Perez, A. J., Ayarzaguena, B., Baldwin, M. P., Dunn Sigouin, E., Furtado, J. C., Garfinkel, C. I., Hitchcock, P., Karpechko, A. Y., Kim, H., Knight, J., Lang, A. L., Lim, E.-P., Marshall, A., Roff, G., Schwartz, C., Simpson, I. R., Son, S.-W., and Taguchi, M.: The role of the stratosphere in subseasonal to seasonal prediction: 1. Predictability of the stratosphere, Journal of Geophysical Research-Atmospheres, https://doi.org/10.1029/2019JD030920, 2020b.

Drouard, M., Rivière, G., Arbogast, P., Drouard, M., Rivière, G., and Arbogast, P.: The North Atlantic Oscillation Response to Large-Scale Atmospheric Anomalies in the Northeastern Pacific, dx.doi.org, 70, 2854–2874, 2013.

Ferranti, L., Corti, S., and Janousek, M.: Flow-Dependent Verification of the ECMWF Ensemble over the Euro-Atlantic Sector, Quarterly Journal of the Royal Meteorological Society, 141, 916–924, https://doi.org/10.1002/qj.2411, 2015.

Garfinkel, C. I., Butler, A. H., Waugh, D. W., Hurwitz, M. M., and Polvani, L. M.: Why might stratospheric sudden warmings occur with similar frequency in El Niño and La Niña winters?, Journal of Geophysical Research, 117, 2012.

[revised manuscript text omitted]

490 Maycock, A. C. and Hitchcock, P.: Do split and displacement sudden stratospheric warmings have different annular mode signatures?, Geophysical Research Letters, 42, 10 943–10 951, 2015.

Maycock, A. C., Masukwedza, G. I. T., Hitchcock, P., and Simpson, I. R.: A Regime Perspective on the North Atlantic Eddy-Driven Jet Response to Sudden Stratospheric Warmings, Journal of Climate, 33, 3901–3917, 2020.

[revised manuscript text omitted]